# Graph Scattering Networks with Adaptive Diffusion Kernel

## Abstract

Scattering networks are deep convolutional architectures that use predefined wavelets for feature extraction and representation. They have proven effective for classification tasks, especially when training data is scarce, where traditional deep learning methods struggle. In this work, we introduce and develop a mathematically sound framework for applying adaptive kernels to diffusion wavelets in graph scattering networks. Stability guarantees with respect to input preturbations are provided. A specific construction of adaptive kernels is presented and applied with continuous diffusion to perform graph classification tasks on benchmark datasets. Our model consistently outperforms traditional graph scattering networks with predefined wavelets, both in scenarios with limited and abundant training data.

## 1 Introduction

Euclidean scattering networks are deep convolutional architectures analogous to Convolutional Neural Networks (CNNs). Unlike standard CNNs, which employ learnable filters at each layer, these networks are equipped with mathematically predefined wavelets selected from a multi-resolution filter bank (Mallat (2012); Bruna & Mallat (2013)). This distinction allows Euclidean scattering networks to serve as mathematically well-understood models that capture the principles underlying the empirical success of CNNs. Specifically, they exhibit proven robustness to small perturbations that are close to translations in the underlying domain (Bruna & Mallat (2013)). In tasks like classification, they are used as feature extractors, requiring only the classifier to be trained. This characteristic makes them particularly advantageous in scenarios with limited data availability, where they deliver state-of-the-art performance while maintaining efficiency comparable to learned deep networks on simpler datasets.

The increasing focus on graph-structured data has spurred interest in adapting CNN architectures to these domains, leading to the development of effective graph convolutional models and variants (e.g. Kipf & Welling (2017), Veličković et al. (2018)). Naturally, proposal on extending the theoretical and practical benefits of Euclidean scattering networks to geometric data follows. Zou & Lerman (2018) first introduced graph scattering networks using spectral wavelets (Hammond et al. (2011), Shuman et al. (2015)) and analyzed its stability with respect to permutations of the nodes and perturbations on the spectrum of the underlying graph domain. Subsequently, Gama et al. (2019b) established improved stability bounds for this family of graph scattering transforms, applicable to more general graphs and independent of their spectral characteristics. Alternatively, Gama et al. (2019a) introduced graph scattering employing diffusion wavelets (Coifman & Maggioni (2006)), using the lazy diffusion operator induced from normalized adjacency, and analyzing stability using diffusion metrics (Nadler et al. (2005), Coifman & Lafon (2006)). Following this, Gao et al. (2019) proposed an alternative graph scattering transform based on lazy random walk diffusion, demonstrating expressivity through extensive empirical evaluations. Furthermore, graph scattering transforms have been extended to spatio-temporal domains (Pan et al. (2021)), pruned to mitigate exponential increase in required resources with network depth (Ioannidis et al. (2020)), and generalized employing functional calculus filters (Koke & Kutyniok (2022)).

A fundamental characteristic shared by all these scattering architectures is the use of fixed, often manually selected filters. Since different diffusion kernels can extract distinct properties from a dataset (Coifman et al. (2005a)), selecting an appropriate kernel is critical to the effectiveness of

a scattering transform. This selection should not be arbitrary; it must ensure the preservation of desirable expressivity and stability properties in the resulting transform. Given that scattering networks are typically used with a trainable classifier, such as a multilayer perceptron (MLP), and the wavelet decomposition operators at each layer are constructed from a single mother wavelet, the fact that only one kernel needs to be learned during classifier training makes this approach particularly promising, as it would likely add only a small number of additional parameters.

In this work, we introduce a mathematically sound framework for incorporating learnable kernels into graph scattering networks. Specifically, we establish stability guarantees for scattering transforms and diffusion wavelets constructed from general diffusion kernels, and propose the design of an adaptive kernel, which is employed in our experiments to demonstrate the enhanced performance of graph scattering with added adaptivity in graph classification tasks. Notably, our stability proofs do not rely on the assumption that the resulting diffusion operator is self-adjoint, a significant relaxation given the spectral theorem. To the best of our knowledge, this is the first work to establish such guarantees. By providing bounds for general diffusion operators, our framework lays the foundation for scattering networks constructed using important operators that are not self-adjoint, as well as for the adaptive designs including such operators.

The paper is organized as follows. Section 2 provides the necessary background. Section 3 discusses the framework for defining diffusion wavelets and metrics (Sec. 3.1) and constructing the diffusion-based graph scattering transform (Sec. 3.2). Section 4 demonstrates the importance of kernel choice with examples of how different kernels capture distinct dataset properties, proposes an adaptive kernel design (Sec. 4.1), establishes energy conservation bounds for wavelets from general kernels (Sec. 4.2), provides a stability analysis of the resulting adaptive diffusion scattering transform (Sec. 4.4), and complexity analysis (Sec. 4.5). Section 5 discusses related work. Section 6 presents numerical results showing that our adaptive graph scattering transform considerably outperforms its non-adaptive counterpart in graph classification tasks across both limited and abundant data settings. It also achieves consistently better performance than other graph learning methods in limited data scenarios, demonstrating its potential for such tasks.

## 2 PRELIMINARIES

We start with some background that will be used for the construction of the adaptive diffusion graph scattering transform (most of which can be found in standard textbooks (e.g. Rudin (1987); Mallat (2008))):

**Metric space:** A metric space is a tuple consisting of a set $X$ and a distance function $d$, which satisfies the metric properties: $\forall x, y, z \in X$: (i) positivity: $d(x, y) > 0$, $\forall x \neq y$; (ii) reflexivity: $d(x, x) = 0$; (iii) symmetry: $d(x, y) = d(y, x)$, and (iv) triangle inequality: $d(x, y) \leq d(x, z) + d(z, y)$. A weighted undirected connected graph $(G, E, W)$, where $W$ assigns positive weights to the edges, is an example of a metric space with the distance between two nodes $x$ and $y$ defined by $d(x, y) = \inf_{p_{x,y}} \sum_{e \in p_{x,y}} w_e$, where $p_{x,y}$ is a path connnecting $x$ and $y$.

**Measure space:** A measure space is a tuple of a set $X$; a $\sigma$-algebra $\Sigma$; and a measure $\mu$ on $(X, \Sigma)$. A $\sigma$-algebra is a nonempty collection of subsets of $X$ closed under set-theoretic operations: complement, countable union, and countable intersection. A **metric measure space** $(X, d, \mu)$ is a measure space with a metric $d$ where the $\sigma$-algebra is induced by the metric, and $\mu$ is a Borel measure.

**Multiresolution analysis**: A multiresolution analysis of $\mathcal{L}^2$ of a metric measure space $(X, d, \mu)$ is a sequence of subspaces $\{V_j\}_{j \in \mathbb{Z}}$, each of which is called an **approximation space**. In the case of $\mathcal{L}^2(\mathbb{R})$, the sequence $\{V_j\}_{j \in \mathbb{Z}}$ satisfies the properties:

   (i) $\lim_{j \to -\infty} V_j = \bigcup_{j=-\infty}^{+\infty} V_j = \mathcal{L}^2(\mathbb{R})$

   (ii) $\lim_{j \to +\infty} V_j = \bigcap_{j=-\infty}^{+\infty} V_j = \{0\}$

   (iii) $V_{j+1} \subseteq V_j$, $\forall j \in \mathbb{Z}$

   (iv) There exists a Riesz basis that spans $V_0$.

The **detail space** $W_j$ is defined as the orthogonal complement of $V_j$ in $V_{j-1}$; in other words, $V_{j-1} = V_j \oplus^\perp W_j$, $\forall j \in \mathbb{Z}$. The orthogonal projection of a signal $x$ on $V_{j-1}$ can thus be decomposed as $P_{V_{j-1}} x = P_{V_j} x + P_{W_j} x$. The projection of a signal $x$ on $W_j$ captures the "details" of $x$ that are

present in the finer-scale space $V_{j-1}$ but absent in the coarser-scale $V_j$. Given a mother wavelet $\psi$, the translations of $\psi$ after being dilated onto scale $2^j$, denoted as $\{\psi_{j,n}\}_{n \in \mathbb{Z}} = \{\frac{1}{\sqrt{2^j}}\psi(\frac{t-2^j n}{2^j})\}_{n \in \mathbb{Z}}$, compose an orthonormal basis of $W_j$. On said basis, the projection of $x$ on $W_j$ can be obtained by a partial expansion: $P_{W_j} x = \sum_{n=-\infty}^{+\infty} \langle x, \psi_{j,n} \rangle \psi_{j,n}$.

**Scattering transform** is a mapping which takes an input signal $x$ and returns a representation $\Phi(x)$, calculated based on a deep convolutional architecture, stable to small deformations while preserves high-frequency information. $\Phi(x)$ is computed by applying sequentially three elements: A **filter bank** of band-pass wavelets $\{\{\psi_{j,k}\}_{k=0}^{K-1}\}_{j=1}^{J}$, a **pointwise nonlinearity** $\rho$ (modulus or ReLU), and an **average operator** $U$. In the Euclidean setting, the filter bank consists of rotated and dilated versions $\psi_{j,k}$ of a mother wavelet $\psi$ with scaling parameter $j$ and angle parameter $k$, with the angle $\theta \in \{2\pi k/K\}_{k=0,\ldots,K-1}$. The scattering representation of $x$ is defined as:

$$\Phi(x) = [S_0(x), S_1(x), \ldots, S_{m-1}(x)] \text{, where}$$
$$S_k(x) = \left[ U\Pi_{i=0}^{k} \left( \rho \psi_{\alpha_i} \right)(x) \right]_{\alpha_0, \alpha_1, \ldots, \alpha_k} \tag{1}$$
$$= \left[ U \left( \rho \left( \ldots \rho \left( \rho \left( x * \psi_{\alpha_0} \right) * \psi_{\alpha_1} \right) \ldots * \psi_{\alpha_k} \right) \right) \right]_{\alpha_0, \alpha_1, \ldots, \alpha_k} .$$

where $\alpha_i$, $i = 0, \ldots, k$ represent the scale parameters.

# 3 GRAPH DIFFUSION SCATTERING TRANSFORM

## 3.1 GRAPH DIFFUSION WAVELETS AND DIFFUSION METRICS

The works in Coifman et al. (2005b); Coifman & Maggioni (2006) introduce a framework for multi-scale and multiresolution analysis on the domain of graphs, based on polyadic powers of a diffusion operator. We consider an undirected, weighted, and connected graph $G = (V, E, W)$, with $|V| = n$ nodes, edges set $E$ and adjacency matrix $W \in \mathbb{R}^{+n \times n}$. The random walk matrix $T = WD^{-1}$ of $G$ defines an induced diffusion process on its nodes, where $D = \text{diag}(d_1, \ldots, d_n)$, and $d_i, i = 1, \ldots, n$ are the degrees of the nodes of $G$. For stability, the lazy diffusion $P = \frac{1}{2}(I + T)$ can be employed. Given that $P$ is left-stochastic and guaranteed to have postive entries at indices $(u, v)$ whenever $(u, v) \in E$, it can also be interpreted as a transition matrix of a random walk process on $G$.

The operator $P$ is mass-preserving (i.e. $\sum_{(u,v) \in E} P[v, u] = 1$ for any fixed $u$), contractive ($||P|| \leq 1$), and positivity-preserving ($x \geq 0 \Rightarrow Px \geq 0$). Consider a random walk on $G$ with $P$ as the transition matrix, the probability distribution starting from an initial $p_0$ (e.g. a Dirac delta $\delta_u$ at any node $u$ of $G$) becomes increasingly "smoothed out" as over time, as observed from the fact that $P^t p_0$ converges to a stationary distribution when $t \to \infty$, and this distribution is independent of $p_0$.

Based on this "smoothening" property, $P$ can be interpreted as a dilation operator, acting on signals on $\mathcal{L}^2(G)$. An analog to the multiresolution analysis can thus be constructed, as proposed in Coifman & Maggioni (2006). In a more general perspective, we consider a diffusion semigroup $\{A^t\}_{t \geq 0}$ induced by a general diffusion operator $A$ acting on $\mathcal{L}^2(X, \mu)$ which satisfies the following properties:

 (i)  $||A^t||_p \leq 1$, for every $1 \leq p \leq +\infty$.
 (ii)  $A^t x \geq 0$, for every $x \geq 0$

Semigroups as such are referred to as Markovian semigroups. We fix a precision level $\epsilon < 1$. Define $A\mathcal{L}^2(X) = \text{span}\{x \in \mathcal{L}^2(X) : ||x|| \leq 1, \frac{||Ax||}{||x||} \geq \epsilon\}$. Let $\lambda_{min} = \inf_{x \in \mathcal{L}^2(X), ||x|| \leq 1} \frac{||Ax||}{||x||}$, and $\lambda_{max} = \sup_{x \in \mathcal{L}^2(X), ||x|| \leq 1} \frac{||Ax||}{||x||}$. As $||A|| \leq 1$, it follows that $\dim(A\mathcal{L}^2(X)) \leq \dim(\mathcal{L}^2(X))$. The operator $A$ contracts the functional space $\mathcal{L}^2(X)$ after each application. The inequality may be strict, as there are signals in some parts of $\mathcal{L}^2(X)$ have their norm contracted by $\lambda_{min}$, which may already be smaller than $\epsilon$.

At times $t_j = \gamma^{j+1}$, where $\gamma > 1$ (commonly set to 2), we discretize $\{A^j\}$ following classical wavelet theory, having wavelets are dilated at scales of polyadic powers. We define the approximation spaces $V_j$ analogous to a multiresolution analysis of $\mathcal{L}^2(X)$ as $A^{t_j}\mathcal{L}^2(X)$. We also convention-ally define $V_{-1} = \mathcal{L}^2(X)$. A family of multiresolution filters, analogous to the wavelets filter bank

in the Euclidean setting, can thus be defined as:

$$\psi_0 = I - A \, , \, \psi_i = A^{t_{i-2}} - A^{t_{i-1}} = A^{2^{i-1}} - A^{2^i} \, (i > 0) \tag{2}$$

These filter can be understood as projecting a signal $x$ onto the complement of $V_j$ in $V_{j+1}$, analogous to the partial expansion of $x$ in the wavelet basis $\{\psi_{j,n}\}$ of $W_j$ (Gama et al. (2019a)), thereby extracting the details of $x$ at coarser scales as $j$ increases.

A diffusion metric can also be constructed on the operator $A$ (Coifman & Lafon (2006)). If $A$ is left-stochastic (i.e. it can be considered as a transition matrix of a Markov chain) and positivity-preserving, then the diffusion distances at time $t$ between two nodes $u$ and $v$ is given by: $d_t(u, v) = ||A^t\delta_u - A^t\delta_v||$. This distance considers all path of length $t$ between $u$ and $v$. If there are many connecting short paths between the two nodes, then $d_t(u, v)$ will be small. It is, as a consequence, robust to noise, unlike the shortest path distance. An additional consequence is that $d_t(u, v)$ is small if $u$'s and $v$'s neighborhoods are similar.

A distance between two graphs of equal sizes can also be defined based on this node-level one, as in Gama et al. (2019a). Given two graphs $G = (V, E, W)$ and $G' = (V', E', W')$ with $|V| = |V'| = n$ and respective diffusion operators $A_G$ and $A_{G'}$, the normalized diffusion distance between $G$ and $G'$ at time $t$ is defined as:

$$d_t(G, G') = \inf_{\Pi \in \Pi_n} ||(A_G^t)^*(A_G^t) - \Pi^{-1}(A_{G'}^t)^*(A_{G'}^t)\Pi|| \tag{3}$$

where $\Pi_n$ is the space of all $n \times n$ permutation matrices, and $A^*$ is the adjoint of operator $A$.

This distance is invariant to node permutation, and is robust to noise similarly to the node-level one. For simplicity, we consider the metric on graphs of equal sizes; however, it can be naturally extended to graphs of different sizes by replacing permutation matrices with soft-correspondences, as in Bronstein et al. (2010) (Gama et al. (2019a)).

## 3.2 GRAPH SCATTERING TRANSFORM

The construction of the multiresolution analysis, and thus an analog of the wavelets filter bank on the domain of graphs, paves the way for the extension of graph scattering transform. Let $\Psi_n : \mathcal{L}^2(X) \to (\mathcal{L}^2(X))^{J_n}$ be the wavelet decomposition operator that maps $x$ to $(\psi_j x)_{j=0,\dots,J_n-1}$, with $\psi_j$ defined as in the previous subsection. Following the Euclidean setting described in Section 2, the diffusion graph scattering transform $\Phi_G(x)$ is also defined from three components: the wavelet decomposition operator at each layer $k$: $\Psi_k$; a pointwise nonlinearity $\rho$; and a low-pass operator $U$. The representation $\Phi(x)$ is calculated analogously to the scattering transform in Equation 1 (see Figure 1).

In Gama et al. (2019a), $\Phi_G(x)$ is introduced with the multiresolution filters being constructed from the intrinsic lazy normalized symmetric adjacency $\overline{P} = \frac{1}{2}(I + M)$ of $G = (V, E, W)$, where $M = D^{-1/2}WD^{-1/2}$. Although $\overline{P}$ is not mass-preserving, there is a spectral theory to this operator. This is desirable in many cases - for example, when constructing a diffusion embedding such that the Euclidean distance in the embedding space corresponds to the diffusion distance in the original graph space (Coifman et al. (2005a)). Moreover, since $\overline{P}$ is contractive

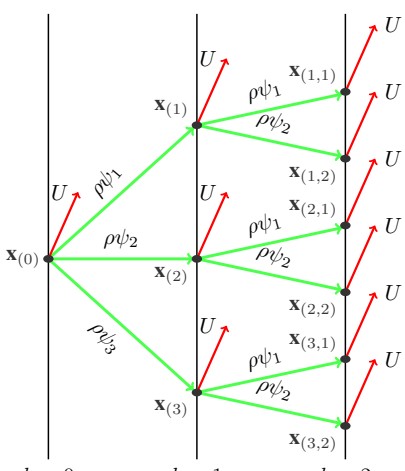

Figure 1: Illustration of graph scattering transform with $m = 3$ layers, scales $J_1 = 3$ and $J_2 = 2$.

(due to its self-adjointness and having spectral radius $\rho(\overline{P}) \le 1$) and positivity preserving, the multiresolution analysis construction remains valid. The average operator $U$ is taken to be the infinite-time diffusion limit $\lim_{t\to\infty} \overline{P}^t$, expressible as $Ux = \langle \mathbf{v}^\mathsf{T}, x \rangle$, where $\mathbf{v} = \frac{\mathbf{d}^{1/2}}{||\mathbf{d}^{1/2}||_2} = \left(\frac{\mathbf{d}}{||\mathbf{d}||_1}\right)^{1/2}$ is the eigenvector of $\overline{P}$ corresponding to the eigenvalue 1, $\mathbf{d}$ being the degree vector of $G$, and $\mathbf{x}^{1/2}$ is the vector with square root of every entry of $\mathbf{x}$.

## 4 ADAPTIVE DIFFUSION KERNELS

In this section, we first discuss examples of how different diffusion kernels can capture different properties of a dataset, as part of our motivation. We then give a design of a learnable kernel that we will use in our experiments as an example to show the enhanced performance. In subsequent parts, we establish stability bounds for general diffusion kernels, which set the ground for other adaptive design that could include other important diffusion operators, which are not self-adjoint.

### 4.1 ADAPTIVE DIFFUSION KERNELS

We consider some examples of $X$ approximately lying on some submanifolds $\mathcal{M}$ of $\mathbb{R}^n$, characterized by a density function $p(x)$, to show how different kernels can captures different properties, e.g. the intrinsic geometry of the data points, its distribution density, or a combination of both (Coifman et al. (2005a)). Between two points $x$, $y$ of $X$, let $k(x, y)$ be the "affinity function" that is symmetric, positivity-preserving, and positive semi-definite. $k(x, y)$ can be interpreted as the analog of edges weight between graph nodes. Let $d(x) = \int_X k(x, y)\mu(y)$ be an analog of degrees of nodes, where $\mu$ is a probability measure. The random walk diffusion operator $A$ can thus be defined as $As(x) = \int_X a(x, y)s(y)d\mu(y)$, where $s$ is a signal on $X$, and $a(x, y) = \frac{k(x,y)}{d(y)}$.

Two examples of diffusion kernels are given in Coifman et al. (2005a), one accounting for the density of the points in $X$, and the other captures the geometry irrespective of density. Consider the random walk diffusion $A_\epsilon$ constructed from an isotropic kernel $k_\epsilon(x, y) = \exp(-||x - y||^2/\epsilon)$. If $p(x)$ is uniform, $A_\epsilon$ approximate the Laplacian-Beltrami operator $\Delta$ on $\mathcal{M}$, as $\epsilon \to 0$ (Belkin & Niyogi (2003)). On the other hand, if $p(x)$ is not, $A_\epsilon$ tends to a more general operator of the form $\Delta + Q$, where $Q(x) = \frac{\Delta p(x)}{p(x)}$ acts as a potential term, reflecting the influence of the non-uniform density.

An alternative normalization is introduced that captures the geometry of the data points by taking into account the non-uniformity of $p(x)$: Let $p_\epsilon(x) = \int_X k_\epsilon(x, y)p(y)dy$, and define the new kernel $\hat{k}_\epsilon(x, y) = k_\epsilon(x, y)/p_\epsilon(x)p_\epsilon(y)$. The corresponding random walk diffusion $\hat{A}_\epsilon$ then serves as an approximation of the Laplace-Beltrami operator at time $\epsilon$, regardless of density variations.

These examples show that the embeddings obtained are highly sensitive to the choice of kernel. Naturally, there are cases where data-driven diffusion is preferable. For each node $u$ of a graph $G$, let the descriptor $g_u$ be a vector that has the characteristics of $u$, e.g. its node degree. Between every two adjacent nodes $u$ and $v$, let $k(u, v)$ be the kernel that quantifies the affinity between the two. For reasons detailed in the next section, we temporarily constraint $k$ to be symmetric. We also require $k$ to be positive on pairs of adjacent nodes for the construction of diffusion filters. Taking inspiration from attentional diffusion in Chamberlain et al. (2021), we propose the kernel between two distinct, adjacent nodes to be given by:

$$k(u, v) = \exp\left(\frac{\langle W(g_u), W(g_w)\rangle}{||W(g_u)||.||W(g_v)||}k_1\right) \tag{4}$$

where $|| \cdot ||$ is the vector norm, $W$ is a mapping from the descriptor space $\mathcal{G}$ to an embedding space $W(\mathcal{G})$, and $k_1$ is a hyperparameter to be tuned. This formulation differs from the kernel used in scale-dot attention (Vaswani et al. (2017)), which is given by $k_{sd}(u, v) = \exp(\frac{(W_K g_u)^\mathsf{T} W_Q g_v}{d_e})$, in two key aspects: First, $k$ is symmetrized by letting $W_K = W_Q$, where both mappings can be nonlinear transformations (e.g. a simple MLP), thereby preserving generalization capability. Second, the inner product is normalized to be the cosine-similarity. In our experiments, we found out that the resulting attention weights without normalization tend to be "extreme", i.e. one neighbor would dominate, causing the attention values to be reduced to either zero or one. As cosine similarity is at most 1, we introduce a relaxing hyperparameter $k_1 \in [0, \infty)$, to extend the possible magnitude range of the kernel. One can apply random walk normalization to $k$ to construct the diffusion kernel. However, as also partially explained in the following section, we reformulate the diffusion kernel $a(u, v)$ between any two nodes (either adjacent or identical) as follows:

$$a(u, v) = \big(k(u, v)/K_u\big) * \big(\sigma(\alpha(u)) * (1 - k_2) + k_2/2\big) \text{ if } u \neq v,$$
$$a(u, u) = 1 - \big(\sigma(\alpha(u)) * (1 - k_2) + k_2/2\big) \tag{5}$$

where $K_u = \sum_{v \in \mathcal{N}(u)} k(u,v)$, $\sigma$ denotes the sigmoid function, $k_2 \in [0,1]$ is an additional hyper-parameter, and $\alpha(u) = \langle W(g_u), \alpha \rangle$, with $\alpha$ is a learnable vector of dimension $\dim(W(\mathcal{G}))$. We introduce $\alpha$ to allow self-diffusion, which is necessary for the convergence of the diffusion operator, to be also learnable. $k_2$ here is used to control the possible range of $a(u,u)$, thereby preventing it becomes too "extreme". We would like to remark that $k_1$ and $k_2$ can be interpreted as regularization hyperparameters, as setting $k_1 = 0$ and $k_2 = 1$ recovers the standard random walk diffusion kernel for the unweighted version of the graph $G$.

Let $\mathbf{A} : \mathcal{G} \to (\mathcal{L}^2(G))^2$ be the operator which maps $g$ to a diffusion matrix $A$ of $g$. By definition, $A$ is left-stochastic. To enhance stability during the training process, we employ a multi-head attention mechanism analogous to that introduced in (Vaswani et al. (2017), Veličković et al. (2018)) by taking the average across the heads: $\mathbf{A}(g) = \frac{\sum_{k=0}^{h-1} \mathbf{A}_k(g)}{h}$.

When strictly discrete time steps are considered, the diffusion matrix can directly be used as the diffusion operator: $A = \mathbf{A}(g)$. However, modeling the diffusion in graph neural networks (GNNs) as a continuous-time process has been shown to enhance both training stability and performance (Wang et al. (2021)). The same approach could thus be done for the above formula. One could discretize update step between two consecutive powers of $A$ by taking fractional temporal difference. Temporal discretization schemes for continuous process can be used for such purpose. A quick discussion of these schemes is given in Appendix A.1. Further experiments are discussed in Section 6 and presented in Appendix A.6.

## 4.2 ADAPTIVE DIFFUSION WAVELETS

The construction of wavelets, in general, relies on the framework of multiresolution analysis. Recall the construction of a multiresolution analysis on graph domains mentioned in Section 3.1. We now prove our formulation of $A$, as presented, is suitable for constructing such an analysis. Unless specified otherwise, all norms are $l^2$-norm.

**Proposition 4.1** *Let $G$ be a connected domain, and $\mathbf{A} : \mathcal{G}^k \to [\mathcal{L}^2(G)]^2$ be the operator that maps descriptor $g$ of the points on $G$ to its diffusion operator $\mathbf{A}(g)$ as above. Define $V_{-1} = \mathcal{L}^2(G)$. Let $A = \mathbf{A}(g)$, and fix a precision $\epsilon > 0$. Given any subspace $V \subseteq \mathcal{L}^2(G)$, denote $A^t V$ be the subspace of $\mathcal{L}^2(G)$ such that $\forall x \in A^t V, \frac{\|A^t x\|}{\|x\|} \geq \epsilon$. The sequence $\{V_j\}_{j \geq -1}$, where $V_j = \mathcal{R}(A^{2^j} V_{-1})$, with $A^{2^j}$ is computed either discretely or continuously, is a multiresolution analysis:*

*(i) $\lim_{j \to +\infty} V_j = \text{span}\{\pi_A\}$, where $\pi_A$ is the unique stationary distribution of the Markov chain induced by $A$.*

*(ii) $V_j \subseteq V_{j-1}$.*

*(iii) There exists a Riesz basis of $V_0$.*

The proof is presented in Appendix A.2. The condition that $A$ has a limiting distribution, which is also the unique stationary distribution of $A$, guarantees the sequence $\{V_j\}$ forms a multiresolution analysis and the dimension of $\lim_{j \to +\infty} V_j$ is minimized. This is obtained by ensuring $A$ is both irreducible (the underlying domain $G$ is connected) and aperiodic ($\exists u : A(u,u) > 0$).

The above multiresolution analysis, constructed using $A$ with the construction of filters in Section 3.1, yields a wavelet decomposition operator $\Psi$ for the proposed adaptive scattering network. It is noteworthy that the formulation of $A$ as defined above encompasses a special family of diffusion operators which is a subset of a broader class, on which diffusion wavelets are constructible. In particular, the diffusion operator $A$ can be any left-stochastic matrix such that for every $x$: $\langle x, \pi_A \rangle = 0$, where $\pi_A$ is any stationary distribution of $A$, and $\|Ax\| < \|x\|$. On these general operators, the resulting wavelet decomposition operator $\Psi$ is proven to be a frame analysis operator, i.e. it defines a frame:

**Proposition 4.2** *On a connected domain $G$, let $\Psi$ be the wavelet decomposition operator on $\mathcal{L}^2(G)$ based on a non-negative matrix $A$ as above: Assume that for every $x \in \mathcal{L}^2(G)$ satisfying $\langle x, \pi_A \rangle = 0, \frac{\|Ax\|}{\|x\|} < 1$. Let $\beta_A = \inf_x (1 - \frac{\|A_G x\|}{\|x\|})$. Then, there exists constants $M(\beta_A), N(\beta_A) > 0$*

*depending only on $\beta_A$ such that for any $x$ as above:*

$$M(\beta_A)||x||^2 \leq \sum_{j=0}^{J-1} ||\psi_j x||^2 \leq N(\beta_A)||x||^2. \tag{6}$$

The proof is presented in Appendix A.3. The existence of the two bounds is a necessary and sufficient condition that there exists a bounded inverse for each decomposition on the image space $\text{Im}(\Psi)$. This means $\Psi$ defines on $\mathcal{L}^2(G)$ a complete and stable representation.

According to the general Perron-Frobenius theory, any irreducible and aperiodic matrix $A$ with non-negative elements has a unique eigenvector $\pi_A$ corresponding to its largest eigenvalue, 1, up to a constant multiple. Furthermore, the remaining eigenvalues of $A$, considered in the unitary space, have strictly smaller moduli. However, there is no guarantee that the orthogonal complements $M_{\pi_A}$ of $\text{span}(\pi_A)$ in $\mathcal{L}^2(G)$ will remain invariant under the action of $A$. As every signal which is a multiple of $\pi_A$ lose all of its information under the wavelet decomposition, to ensure stability, we would want to design $A$ such that $AM_{\pi_A} \subseteq M_{\pi_A}$. A straightforward family of matrices satisfying this property is the class of self-adjoint matrices. Ensuring symmetry in the kernel $k$, as in our construction, is a sufficient condition for this.

It is also worth to noting that the condition that $G$ be connected can be relaxed. Specifically, $G$ can consist of $p$ connected components that are pairwise disconnected, provided $p \ll |V| = n$. This condition is necessary because each component can have its own stationary distribution, making the subspace of stationary distributions of $A$ on $\mathcal{L}^2(G)$ of multiple dimensions, with a maximum dimension of $p$. Any signal in this subspace will lose all of its information upon applying $\Psi$, thus rendering $\Psi$ useless for such signals. For simplicity, we continue to consider the case where $G$ has only 1 connected component.

### 4.3 ADAPTIVE GRAPH SCATTERING TRANSFORM

We construct our Adaptive Graph Scattering Transform similarly to the one in Section 3.2 by replacing the non-adaptive decomposition operator with the adaptive version defined in Section 4.2. Additionally, we employ the average mean pooling operator $U$, which is independent of $A$: $Ux = \langle 1/\mathbf{n}, x \rangle$. In particular, on a connected graph $G$ with a graph signal $x$, the transformation at each layer is given by:

$$\begin{aligned} \phi_k = U(\rho\Psi)^k x &= \left[U\Pi_{i=0}^k \left(\rho\psi_{j_i}\right)(x)\right]_{j_0,j_1,\ldots,j_k} \\ &= \left[U\rho\psi_{j_k} \ldots \rho\psi_{j_1}\rho\psi_{j_0}x\right]_{j_0,j_1,\ldots,j_k}. \end{aligned} \tag{7}$$

where $\{\psi_{j_i}\}_{j_i}$ are multiresolution filters constructed using the adaptive operator $A$. Thus, the scattering representation obtained from an $m$-layer network is:

$$\Phi(x) = [Ux, \phi_1(x), \ldots, \phi_{m-1}(x)] = \left[Ux, U\rho\Psi x, \ldots, U(\rho\Psi)^{m-1}x\right] \tag{8}$$

In the following we provide the stability analysis of the adaptive graph scattering transforms using general diffusion kernel:

### 4.4 STABILITY ANALYSIS

A robust and meaningful signal representation should exhibit stability to noise, meaning that a small change in the input signal yields proportionally small variations in the output representation. As mentioned in Section 3.1, the matrix $A$, being both positive-preserving and contractive, naturally induces a graph-level diffusion metric. We begin by establishing the stability of the wavelet decomposition operator with respect to this graph metric.

**Lemma 4.1** *On two distinct graphs $G$ and $G'$, let $\Psi_G$ and $\Psi_{G'}$ be the wavelet decomposition operators induced from respectively $A_G$ and $A_{G'}$. For a signal $x$ both orthogonal to limiting distributions $\pi_{A_G}$ of $A_G$ and $\pi_{A_{G'}}$ of $A_{G'}$, let $\beta = \min\left(\inf_x\left(1 - \frac{||A_G x||}{||x||}\right), \inf_x\left(1 - \frac{||A_{G'} x||}{||x||}\right)\right)$. We have:*

$$\inf_{\Pi \in \Pi_n} ||\Psi_G - \Pi\Psi_{G'}\Pi^\mathsf{T}|| \leq 2\sqrt{2}\sqrt{\frac{(1-\beta)^2(1-\beta+\beta^2)}{(2\beta-\beta^2)^3}}d(G,G') \tag{9}$$

*where $d(G, G')$ is the diffusion metric between two graphs mentioned in Section 3.1.*

The complete proof is presented in Appendix A.4. This lemma serves as the primary tool in our proof of the next result, which establishes stability bounds for an $m$-layer graph scattering network under small perturbations to the graph structure:

**Theorem 4.1** *Let $x \in \mathbb{R}^n$ and $\Phi_G(x)$ be the $m$-layer scattering representation of a signal $x$ on a graph $G$, and let $\Phi_{G'}(x)$ be the same respectively on graph $G'$. Let $\beta = \min\left(\inf_x \left(1 - \frac{||A_G x||}{||x||}\right), \inf_x \left(1 - \frac{||A_{G'} x||}{||x||}\right)\right)$ and $N(\beta)$ be as in Proposition 4.2. We have:*

$$||\Phi_G(x) - \Phi_{G'}(x)||^2 \leq \sum_{k=0}^{m-1} \left[ kN(\beta)^{k-1} \sqrt{\frac{8(1-\beta)^2(1-\beta+\beta^2)}{(2\beta-\beta^2)^3}} d(G, G') \right]^2 ||x||^2 \qquad (10)$$

The proof is presented in details in Appendix A.5. Theorem 4.1 gives the stability bound for the scattering representations of the same signal $x$ on two different graphs $G$ and $G'$. Each graph has its own multiresolution analysis, and if the distance between the two graph is small, then the discrepancy between the resulting representations will also be small. Although $||\Phi_G - \Phi_{G'}||$ depends exponentially on $m$, it is important to note that, in most applications, $m$ typically is smaller than 5. The change in scattering representations given a small topological preturbation can thus be effectively characterized by a linear dependence on $d(G, G')$.

### 4.5 COMPLEXITY

**Number of parameters:** In this work, we adopt the traditional architecture of the scattering transform, where the same wavelet decomposition operator is used throughout, and all of its wavelets are generated from a single mother wavelet. Consequently, only one filter needs to be learned across the entire scattering network. The additional number of parameters compared to traditional scattering is $\mathcal{O}(KPH)$, where $K$ is the size of each descriptor, $P$ is the number of parameters in the mapping $W$, and $H$ is the number of heads. This does not depend on the size of the scattering network or the size of the graph $G$.

**Memory requirement:** We consider a scattering network of $m$ layers, and each layer has $k$ wavelets. Since the model has to store the attributes in each wavelet scale for doing low-pass averaging and diffusion in subsequent layer, the memory requirement is $\mathcal{O}(Ck^m N)$, where $C$ is the number of input channels, $N = |V|$ is the number of graph nodes. Since $m$ and $k$ are predefined hyperparameters, with $m \leq 5$ in most applications (as scattering energy rapidly diminishes in deeper layers with increasing $m$ (Bruna & Mallat (2013))), the memory requirement effectively scales linearly with the number of graph nodes.

## 5 RELATED WORKS

The incorporation of adaptivity directly into graph scattering networks has been explored in prior works, but only to a limited extent. Recently, Tong et al. (2024) introduced a scale-adaptive extension of the lazy random walk diffusion scattering transform, enabling adaptive wavelet scale adjustment. Their approach demonstrated competitive performance compared to popular GNNs and the original graph scattering network. With a different perspective, Wenkel et al. (2022) proposed a hybrid GNN combining the low-pass filter of graph convolutional networks (GCNs) with the band-pass filter of graph scattering networks at each layer to capture multi-scale information. For Euclidean scattering, Oyallon et al. (2018) integrated scattering networks with deep residual networks (He et al. (2016)) to achieve comparable image classification results with fewer layers. Additionally, Zarka et al. (2021) introduced a scattering model with only $1 \times 1$ convolutional tight frames for scattering feature projection, delivering similar performance.

## 6 NUMERICAL EXPERIMENTS

In this section, we empirically demonstrate the discriminative power of the adaptive diffusion scattering transform in classification tasks on two types of datasets: social networks and bioinformatics.

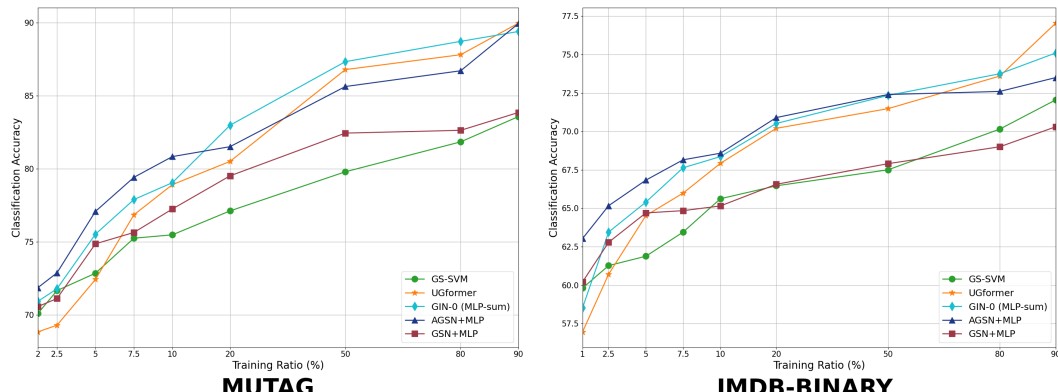

Figure 2: Classification accuracies as a function of percentage of training data used in the bioinformatics dataset (MUTAG) and the social network dataset (IMDB-BINARY)

We perform graph-level classification on well-known social network and bioinformatics datasets as described in Morris et al. (2020). To maintain consistency with our theoretical framework, we restrict our experiments to datasets comprising connected graphs, namely, COLLAB, IMDB-BINARY, IMDB-MULTI, and MUTAG (Morris et al. (2020)). A detailed description of these datasets is provided in Appendix A.7. For each node $u$, the descriptor $g_u$ is chosen to be a vector consists of topological features of $u$ and its neighborhood: degree, eccentricity, clustering coefficient, number of triangles contains $u$ as a vertex, core number, clique number, and PageRank. The bioinformatics dataset MUTAG includes 7 node features, which we directly use as input to the diffusion process. Conversely, the social network datasets lack inherent node features; therefore, we use the descriptors as a proxy.

For the classification task, we employ a model that integrates our adaptive graph scattering network as a feature extractor with a simple MLP as the classifier, denoted as AGSN+MLP. Using the MLP enables backpropagation, thereby facilitating the learning of the kernel weights in our model. For comparison, we implement the same architecture but with a lazy random walk kernel $\frac{1}{2}(I + WD^{-1})$, referred to as GSN+MLP.

**Performance:** We evaluate the performance of AGSN+MLP against traditional graph scattering methods (GSN+MLP, GS-SVM Gao et al. (2019)), graph transformer (UGformer (Nguyen et al. (2022))), and graph neural network (GIN-0 (MLP-sum) (Xu et al. (2018a))) on smaller datasets with varying training data sizes. These models were chosen for their publicly available implementations. Figure 2 shows the classification accuracy as a function of the percentage of samples used for training, with the remaining data reserved for validation. Further experimental details are provided in Appendix A.7. When training data is scarce ($\leq 5\%$), graph deep learning models deteriorate rapidly, with UGformer showing a particularly steep decline. In contrast, graph scattering methods show a more stable performance drop and outperform deep learning models under limited data. AGSN+MLP excels in these scenarios, maintaining a clear advantage over both model types, particularly where graph deep learning deteriorates, and graph scattering begins to excel. This is due to its ability to inherit the predefined structure of the graph scattering transform while controllably adapting the diffusion process to the task. The performance gap is more pronounced on IMDB-BINARY, compared to MUTAG's smaller size (188 samples), where extremely limited data ($\sim$3 samples for 2%) reduces differentiation among models. Conversely, with abundant training data ($\geq 80\%$), graph deep learning models (UGformer, GIN-0) outperform most scattering methods.

**Running time:** We compare the total end-to-end running time of the five models using 2.5% of IMDB-BINARY as training data on the same NVIDIA A100 40GB GPU, shown in Figure 3 (logarithmic scale). For this dataset, GIN-0, UGformer, and AGSN+MLP require adding node features, while GS-SVM and GSN+MLP additionally require extracting the scattering representation. Neural network models are trained and validated for 200 epochs, whereas GS-SVM is fitted only once. AGSN+MLP is a little bit slower (roughly 3.5x) than GSN+MLP and GIN-0 since the gradients are also backpropagated through the scattering architecture, but AGSN+MLP achieves significantly higher accuracy than the other models.

**Additional experiments:**

As mentioned in Section 4.1, the diffusion process can be modeled as a continuous one. In Appendix A.6, we present additional experiments showing the choice of time step or temporal discretization schemes largely affect the stability of the training process. Increasing the time step or employing discretization schemes with higher numerical accuracy improves the numerical precision of each weight update, resulting in a more stable and refined training curve, similarly to adjusting the learning rate. However, due to the highly non-convex nature of the optimization problem in our case, this does not necessarily translate to better performance as observed in Wang et al. (2021) for linear GCNs. A balance should be achieved between stability and the ability to escape local minima. Consequently, we treat the time step as a hyperparameter in our experiments.

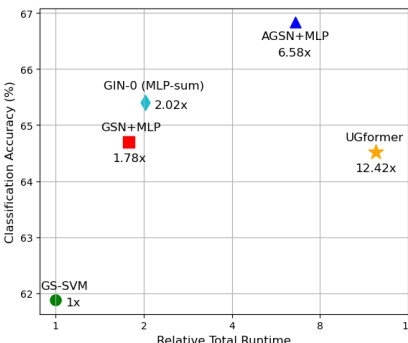

Figure 3: Total running time versus classification accuracy on IMDB-BINARY with $2.5\%$ data for training.

To compare AGSN+MLP with other models under abundant training data, we perform 10-fold cross-validation using 90% of the data for training and evaluate classification accuracies across four datasets. Results from other models are taken from their original papers and detailed in Appendix A.6. In these scenarios, AGSN+MLP outperforms most graph kernel methods but shows lower performance compared to graph deep learning models. Compared to other graph scattering methods, AGSN+MLP performs better than GS-SVM and is comparable to GGSN+EK (Koke & Kutyniok (2022)) on smaller datasets (MUTAG, IMDB-BINARY). However, its performance shows negligible differences or is slightly lower on larger datasets (IMDB-MULTI, COLLAB). Due to slight differences in the cross-validation procedure used in this work and that described in Koke & Kutyniok (2022), we believe additional experiments are necessary for a fairer comparison. Unfortunately, the original implementation of GGSN+EK is not publicly available.

Compared to graph deep learning models, we attribute AGSN+MLP's lower performance in abundant training data scenarios to limitations in utilizing the scattering representation. In image processing, Zarka et al. (2020) demonstrated that incorporating sparse $l^1$ dictionary coding into the scattering architecture to reduce intra-class variability, before inputting it into the classifier, achieves performance comparable to AlexNet (Krizhevsky et al. (2012)) on the ImageNet dataset (Deng et al. (2009)). Adapting similar strategies for graph structures is also an interesting direction for future research.

## 7 CONCLUSIONS

In this work, we developed a mathematically sound framework for the application of learnable kernels in graph scattering networks, allowing for data-driven feature extraction through adaptive diffusion. We established the stability of this adaptive scattering under both small signal and topological perturbations in the underlying graph domain. In particular, we show that the distance between two scattering representations of a graph signal on two different graphs is proportional to the diffusion distance between the graphs, based on general diffusion operators, setting the ground for adaptive designs containing important operators that are not self-adjoint. Additionally, we empirically showed on social network and bioinformatics datasets that the application of adaptive kernels to scattering network improves performance considerably, in both scenarios of limited and abundant training data.

Our results open up several promising research directions. One potential direction is to explore more designs of general adaptive kernels that are not restricted to being self-adjoint. Since the diffusion metric depends on $A$, it would also be beneficial to refine the stability bounds so that they rely solely on the topological properties of the graph.

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

# A APPENDIX

## A.1 DISCUSSION ON TEMPORAL DISCRETIZATION SCHEMES

Temporal discretization schemes for continuous diffusion processes are broadly classified into two categories: explicit and implicit. These schemes can further be categorized into single-step and multi-step.

The simplest form of temporal discretization of a continuous feature update process is done by replacing the continuous-time derivative with forward time difference:

$$\frac{x_i^{(k+1)} - x_i^{(k)}}{\tau} = \sum_{j:(i,j)\in E} a(g_j, g_i)x_j^{(k)} - a(g_i, g_j)x_i^{(k)} = \left(\sum_{j:(i,j)\in E} a(g_j, g_i)x_j^{(k)}\right) - x_i^{(k)} \quad (11)$$

where $k$ is the discrete time index, and $\tau$ is the time step. Rewriting this in matrix form yields the explicit Euler scheme:

$$x^{(k+1)} = \tau(A - I)x^{(k)} + x^{(k)} = \tau\bar{A}x^{(k)} + x^{(k)} = (I + \tau\bar{A})x^{(k)} \quad (12)$$

Implicit schemes, instead, employ a backward temporal difference:

$$\frac{x_i^{(k+1)} - x_i^{(k)}}{\tau} = \sum_{j:(i,j)\in E} a(g_j, g_i)x_j^{(k+1)} - a(g_i, g_j)x_i^{(k+1)} = \left(\sum_{j:(i,j)\in E} a(g_j, g_i)x_j^{(k+1)}\right) - x_i^{(k+1)} \quad (13)$$

which, in matrix form, becomes:

$$(I - \tau\bar{A})x^{(k+1)} = x^{(k)} \quad (14)$$

To compute the update $x^{(k+1)}$ from $x^{(k)}$, one have to solve a linear system, hence this approach is called implicit.

A multi-step scheme, for higher numerical accuracy, uses intermediate fractional time steps for updating $x^{(k+1)}$. One of the most widely used multi-step methods is the Runge-Kutta family, with the fourth-order Runge-Kutta (RK4) scheme being the most prominent. The subsequent iterate in RK4 is calculated as:

$$x^{(k+1)} = x^{(k)} + \frac{1}{6}(k_1 + 2k_2 + 2k_3 + k_4), \quad (15)$$

where:

$$\begin{aligned} k_1 &= \tau\bar{A}x^{(k)} & k_3 &= \tau\bar{A}(x^{(k)} + k_2/2) \\ k_2 &= \tau\bar{A}(x^{(k)} + k_1/2) & k_4 &= \tau\bar{A}(x^{(k)} + k_3) \end{aligned} \quad (16)$$

In general, linear $k$-step methods are of the form:

$$\sum_{j=0}^{s} \alpha_j x^{(k+j)} = \tau \sum_{j=0}^{s} \beta_j \bar{A}x^{(k+j)} \quad (17)$$

These methods achieve increased accuracy by increasing the number of calculations. In the case of Runge-Kutta methods, RK4 strikes a balance between the two, offering sufficient precision without excessive overhead (Butcher (2016)). As with Euler's method, how these are implicit or explicit depends on the constants $\alpha_j$ and $\beta_j$.

## A.2 PROOF FOR PROPOSITION 4.1

**Proof:**

We recall here the formula of $A$:

$$A(u, v) = \left(\text{softmax}_{v\in\mathcal{N}(v)}(k(u, v))\right) * \left(\sigma(\alpha) * (1 - k_2) + \frac{k_2}{2}\right) \text{ if } u \neq v, (u, v) \in E$$

$$A(u, u) = 1 - \left(\sigma(\alpha) * (1 - k_2) + \frac{k_2}{2}\right) > 0$$

where $k(u, v) = \exp\left(\frac{\langle W(g_u), W(g_w)\rangle}{||W(g_u)|| ||W(g_v)||} k_1\right)$.

(i) Given that $A = \mathbf{A}(g)$ is aperiodic (i.e., $\exists u : A(u, u) > 0$) and irreducible (the domain $G$ is connected), it follows from the theory of time-homogeneous Markov chains that $A$ admits a unique stationary distribution, denoted as $\pi_A$. Hence, for any $x \in \mathcal{L}^2(G)$, the following limit holds:

$$\forall x \in \mathcal{L}^2(G), \lim_{t \to \infty} A^t x = k\pi_A,$$

where $k$ is a scalar depending only on the initial condition $x$. Consequently, we have $\lim_{j \to +\infty} V_j = \text{span}\{\pi_A\}$.

(ii) As the underlying kernel $k$ of $A$ is symmetric, $A$ is a self-adjoint operator. Furthermore, since any temporal discretization of $A$ is represented as a sum of scalar multiplications of $A$, the resulting diffusion operator preserves the self-adjointness. Let $\lambda_1 = 1$, $\lambda_2, \ldots, \lambda_n$ denote the eigenvalues of $A$ arranged in non-increasing order, and let $x_1, x_2, \ldots, x_n$ be the corresponding eigenvectors. By the iterative definition $V_j = AV_{j-1}$, and since $0 < \lambda_n < 1$, it follows that $V_j \subseteq V_{j-1}$.

(iii) Consider the eigenvectors $x_i$ corresponding to eigenvalues $\lambda_i$ such that $\lambda_i > \epsilon$. Due to the self-adjointness of $A$, these eigenvectors are orthogonal, and hence they form a Riesz basis for the initial subspace $V_0 = AV_{-1}$.

### A.3 Proof for Proposition 4.2

**Proof:** Let $S = \sum_{j=0}^{J-1} ||\psi_j x||^2$.

In the case of the lower bound, we have:

$$S = \sum_{j=0}^{J-1} ||\psi_j x||^2 = ||I - A||^2 + ||A - A^2||^2 + \ldots + ||A^{2^{J-1}} - A^{2^J}||^2$$

$$\geq ||I - A||^2 \geq (1 - (1 - \beta_A))^2 = \beta_A^2 = M(\beta_A)$$

For the upper bound, we consider the complexification $\mathcal{L}^2(G)^+$ of our functional space $\mathcal{L}^2(G)$. $\mathcal{L}^2(G)^+$ consists of pairs of vectors $(x, y)$ in $(\mathcal{L}^2(G))^2$, where addition is defined by $(x_1, y_1) + (x_2, y_2) = (x_1 + x_2, y_1 + y_2)$ and scalar multiplication with any complex number is given by $(a + bi)(x, y) = (ax - by, bx + ay)$, similarly to the case if we consider $(x, y) = x + iy$. A linear transformation $A$ on $\mathcal{L}^2(G)$ is extended to its complexification $A^+$ on $\mathcal{L}^2(G)^+$ by defining: $A^+(x, y) = Ax + iAy$. A simple observation is that the real vector space is a special case of its complexification, where $y = 0$.

As every linear transformation of dimension $n$ in a unitary vector space has fully $n$ eigenvalues, we consider $S$ as a function of the eigenvalues of $A^+$. Let $Q_J(x) = ||1 - \lambda||^2 + \sum_{j=1}^{J-1} ||\lambda^{2^{j-1}} - \lambda^{2^j}||^2$, and $\alpha_A = 1 - \beta_A$. The upper bound of $S$ is then the upper bound of $Q_J(x)$, where $0 \leq |\lambda| \leq \alpha_A$. Let $a = \Re(\lambda)$ and $b = \Im(\lambda)$. Let $Q(x) = ||1 - \lambda||^2 + \sum_{j=1}^{\infty} ||\lambda^{2^{j-1}} - \lambda^{2^j}||^2$. We notice that:

$$Q(\lambda^2) = Q(\lambda) + ||1 - \lambda^2||^2 - ||1 - \lambda||^2 - ||\lambda||^2 ||1 - \lambda||^2 = Q(\lambda) + 2a||1 - \lambda||^2$$

When $a \geq 0$, $Q(\lambda^2) \geq Q(\lambda)$. Therefore $Q_J(\lambda) \leq \sup_{||\lambda|| \leq \alpha_A, \Re(\lambda) \geq 0}(Q(\lambda)) = \lim_{||\lambda|| \to 0} Q(\lambda) = Q(0) = 1$.

On the other hand, when $a < 0$, $Q(\lambda^2) \leq Q(\lambda)$. Let $\alpha_\lambda = |\lambda| \in [\alpha_A^2, \alpha_A]$. Then, for any fixed $\alpha_\lambda$:

$$Q(\lambda) = |1 - \lambda|^2 + |1 - \lambda|^2 \alpha_\lambda^2 + \ldots + (\alpha_\lambda^{2^{j-1}})^2 |1 - \lambda^{2^{j-1}}|^2 + \ldots$$

$$\leq |1 + \alpha_\lambda|^2 + |1 + \alpha_\lambda|^2 \alpha_\lambda^2 + \ldots (\alpha_\lambda^{2^{j-1}})^2 |1 + \alpha_\lambda^{2^{j-1}}|^2 + \ldots$$

$$= 1 + 2\alpha_\lambda + \alpha_\lambda^2 + \alpha_\lambda^2 + 2\alpha_\lambda^3 + \alpha_\lambda^4 + \ldots + \alpha_\lambda^{2^j} + 2\alpha_\lambda^{2^{j-1}+2^j} + \alpha_\lambda^{2^{j+1}} + \ldots$$

$$= 1 + 2(\alpha_\lambda + \alpha_\lambda^2 + \alpha_\lambda^3 + \alpha_\lambda^4 + \ldots + \alpha_\lambda^{2^j} + \alpha_\lambda^{2^{j-1}+2^j} + \ldots)$$

As it is straight-forward that this sequence is absolutely convergent for $0 \leq \alpha_\lambda < 1$, we thus have:

$$Q_J(\lambda) \leq \sup_{\alpha_A^2 \leq ||\lambda|| \leq \alpha_A, \Re(\lambda)<0} Q(\lambda) \leq 1 + 2(\alpha_A + \alpha_A^2 + \alpha_A^3 + \alpha_A^4 + ... + \alpha_A^{2^j} + \alpha_A^{2^{j-1}+2^j} + ...) = N(\beta_A).$$

Therefore, $S \leq N(\beta_A)$, where $N(\beta_A)$ is defined as above.

## A.4 Proof for Lemma 4.1

This proof follows partly the strategy used in Gama et al. (2019a).

**Proof:** We have, for any signal $x$ the orthogonal complement of $\text{span}\{\pi_{A_G}, \pi_{A_{G'}}\}$:

$$||\Psi_G - \Psi_{G'}||^2 = \left|\left| \begin{bmatrix} \psi_0(G) \\ \psi_1(G) \\ \vdots \\ \psi_{J-1}(G) \end{bmatrix} - \begin{bmatrix} \psi_0(G') \\ \psi_1(G') \\ \vdots \\ \psi_{J-1}(G') \end{bmatrix} \right|\right|^2 = \sum_{j=0}^{J-1} ||\psi_j(G) - \psi_j(G')||^2$$

Since $\psi_j(G) = A_G^{2^{j-1}} - A_{G'}^{2^j}$,

$$\begin{aligned} ||\Psi_G - \Psi_{G'}||^2 &\leq \sum_{j=0}^{J-1} 2\left( ||A_G^{2^{j-1}} - A_{G'}^{2^{j-1}}||^2 + ||A_G^{2^j} - A_{G'}^{2^j}||^2 \right) \\ &\leq \sum_{j=0}^{J-1} 4||A_G^{2^j} - A_{G'}^{2^j}||^2 \end{aligned} \tag{18}$$

by triangle inequality.

For any $n \in \mathbb{N}$ and any two matrices $M, N$ with norm strictly less than 1, let $\lambda = \max(||M||, ||N||)$, Let $g(t) = (tM + (1-t)N)^n$. We have:

$$||M^n - N^n|| = ||g(0) - g(1)|| = \left|\left| \int_0^1 g'(t)dt \right|\right| \leq \int_0^1 ||g'(t)||dt \leq \sup_{t \in (0,1)} ||g'(t)||.$$

Since $g'(t) = \sum_{i=0} n(tM + (1-t)N)^i(M-N)(tM + (1-t)N)^{n-(i+1)}$, we thus have $||M^n - N^n|| \leq ||g(t)|| \leq \sum_{i=0} n\beta^i ||M-N||\beta^{n-(i+1)} = n\beta^{n-1}||M-N||$

Applying this to equation 18:

$$\begin{aligned} ||\Psi_G - \Psi_{G'}||^2 &\leq 4\sum_{j=0}^{J-1} 2^{2j}(1-\beta)^{2^{j+1}}||A_G - A_{G'}||^2 \\ &\leq 4\sum_{t=0}^{2^{J-1}} t^2(1-\beta)^{2t}||A_G - A_{G'}||^2 \end{aligned}$$

Let $h(x) = \sum_{t=0}^{\infty} t^2 a^{xt}$ and $H(x) = \int \left( \int h(x)dx \right) dx$ for a constant $a$. Then $h(2) = \frac{d}{dx^2}H(2) = \frac{1}{(\ln(a))^2}\frac{d}{dx^2}\left(\frac{1}{1-a^x}\right) = \frac{a^2(1+a^2)}{(1-a^2)^3}$. Thus,

$$\begin{aligned} ||\Psi_G - \Psi_{G'}||^2 &\leq 4||A_G - A_{G'}||^2 \frac{(1-\beta)^2(1+(1-\beta)^2)}{(1-(1-\beta)^2)^3} \\ &= 4||A_G - A_{G'}||^2 \frac{2(1-\beta)^2(1-\beta+\beta^2)}{(2\beta-\beta^2)^3} \end{aligned}$$

Therefore,

$$||\Psi_G - \Psi_{G'}|| \leq 2\sqrt{2}||A_G - A_{G'}||\sqrt{\frac{(1-\beta)^2(1-\beta+\beta^2)}{(2\beta-\beta^2)^3}}$$

As this inequality stays true for any node permutation on $G'$, lemma 4.1 is thus proved.

## A.5 PROOF FOR THEOREM 4.1

**Proof:**

Let $\epsilon = \sqrt{\frac{8(1-\beta)^2(1-\beta+\beta^2)}{(2\beta-\beta^2)^3}} d(G, G')$.

The case $k = 0$ is straight-forward as $U$ is independent of $G$ and $G'$. For the case $k = 1$:

$$||U\rho\Psi_G - U\rho\Psi_{G'}|| \leq ||U\rho(\Psi_G - \Psi_{G'})|| \leq \epsilon$$

For general $k \geq 2$:

$$
\begin{aligned}
||U(\rho\Psi_G)^k - U(\rho\Psi_{G'})^k|| &= ||U(\rho\Psi_G(\rho\Psi_G)^{k-1} - \rho\Psi_G(\rho\Psi_{G'})^{k-1} + \rho\Psi_G(\rho\Psi_{G'})^{k-1} \\
&\quad - \rho\Psi_{G'}(\rho\Psi_{G'})^{k-1})|| \\
&\leq ||U(\rho\Psi_G((\rho\Psi_G)^{k-1} - (\rho\Psi_{G'})^{k-1}) + (\rho\Psi_G - \rho\Psi_{G'})(\rho\Psi_{G'})^{k-1})|| \\
&\leq N(\beta)(k-1)(N(\beta))^{k-2}\epsilon + \epsilon(N(\beta))^{k-1} \\
&= k(N(\beta))^{k-1}\epsilon
\end{aligned}
$$

We thus have, by induction, $||U(\rho\Psi_G)^k - U(\rho\Psi_{G'})^k|| \leq k(N(\beta))^{k-1}\epsilon$.

Since,

$$||\Phi_G(x)||^2 = \sum_{k=0}^{m-1} ||U(\rho\Psi_G)^k(x)||^2$$

thus

$$||\Phi_G - \Phi_{G'}||^2 = \sum_{k=0}^{m-1} ||U(\rho\Psi_G)^k - U(\rho\Psi_{G'})^k||^2 \leq \sum_{k=0}^{m-1} k(N(\beta))^{k-1}\epsilon .$$

## A.6 ADDITIONAL EXPERIMENTS

**Effect of time step when using fractional temporal difference in training:**

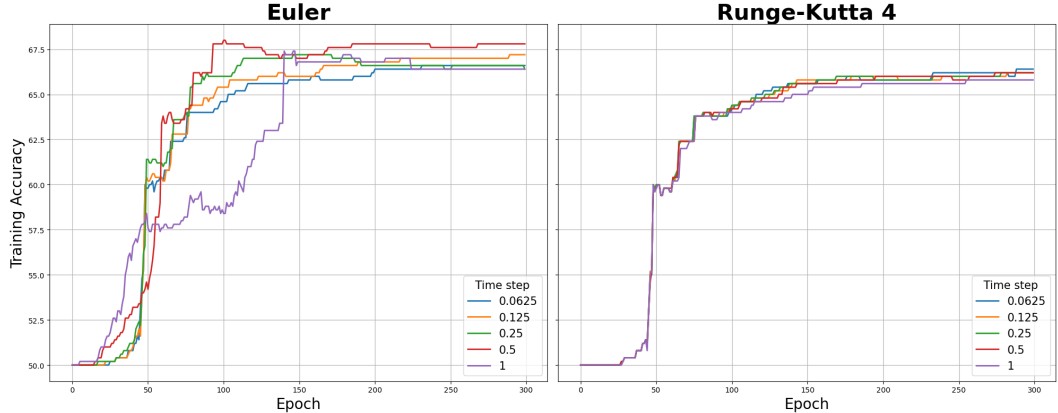

Figure 4: Training accuracies as a function of epoch for different time step and approximation scheme choices on IMDB-BINARY with otherwise same configuration and initialization.

**10-folds cross-validation results on COLLAB, IMDB-BINARY, IMDB-MULTI, and MUTAG:**

To facilitate comparison with other types of models, we perform 10-fold cross-validation using 90% of data for training, and compare the classification accuracies to other graph deep learning, scattering, and kernel methods. These are Weisfeiler-Lehman kernel (WL) (Shervashidze et al. (2011)), Graphlet kernels (Shervashidze et al. (2009)), DGK (Yanardag & Vishwanathan (2015)), GS-SVM (Gao et al. (2019)), GGSN+EK (Koke & Kutyniok (2022)), DGCNN (Zhang et al. (2018)),

| Methods | Accuracies (%) | | | |
|---|---|---|---|---|
| | COLLAB | IMDB-BINARY | IMDB-MULTI | MUTAG |
| WL | $77.82 \pm 1.45$ | $71.60 \pm 5.16$ | N/A | $84.11 \pm 1.91$ |
| Graphlet | $73.42 \pm 2.43$ | $65.4 \pm 5.95$ | N/A | $85.2 \pm 0.9$ |
| DGK | $73.0 \pm 0.2$ | $66.9 \pm 0.5$ | $44.5 \pm 0.5$ | $87.4 \pm 2.7$ |
| DGCNN | $73.76 \pm 0.49$ | $70.03 \pm 0.86$ | $47.83 \pm 0.85$ | $85.83 \pm 1.66$ |
| PSCN | $72.6 \pm 2.15$ | $71.00 \pm 2.29$ | $45.23 \pm 2.84$ | $88.95 \pm 4.37$ |
| S2S-N2N-PP | $81.75 \pm 0.8$ | $73.8 \pm 0.7$ | $51.19 \pm 0.5$ | $89.86 \pm 1.1$ |
| GSN-e | $85.5 \pm 1.2$ | $77.8 \pm 3.3$ | $54.3 \pm 3.3$ | $90.6 \pm 7.5$ |
| GIN-0 (MLP-sum) | $80.20 \pm 1.9$ | $75.10 \pm 5.10$ | $52.30 \pm 2.80$ | $89.40 \pm 5.60$ |
| GS-SVM | $79.94 \pm 1.61$ | $71.20 \pm 3.25$ | $48.73 \pm 2.32$ | $83.57 \pm 6.75$ |
| GGSN+EK | $80.34 \pm 1.68$ | $73.20 \pm 3.76$ | $49.47 \pm 2.27$ | N/A |
| GSN+MLP | $75.2 \pm 1.31$ | $71.3 \pm 3.94$ | $45.11 \pm 3.13$ | $83.85 \pm 4.74$ |
| **AGSN+MLP** | $77.25 \pm 0.99$ | $73.5 \pm 4.48$ | $48.54 \pm 4.5$ | $89.94 \pm 5.74$ |

Table 1: Classification Accuracies on Social Network and Bioinformatics datasets. The first, second, and third best performing models are respectively highlighted in green, yellow, and orange.

PSCN (Niepert et al. (2016)), S2S-N2N-PP (Jin & JaJa (2018)), GSN-e (Bouritsas et al. (2022)), and GIN-0 (MLP-sum) (Xu et al. (2018b)). The results are obtained from the corresponding original publications. Detailed information regarding the experimental setup is presented in Appendix A.7. The classification accuracies are summarized in Table 1.

## A.7 DATASETS AND EXPERIMENTS SET-UP

We provide here additional information on datasets and training procedures.

**Datasets:**

| Attributes | COLLAB | IMDB-BINARY | IMDB-MULTI | MUTAG |
|---|---|---|---|---|
| Num. of graphs | 5000 | 1000 | 1500 | 188 |
| Avg. nodes | 74.5 | 19.8 | 13.0 | 17.9 |
| Avg. edges | 2457.8 | 96.5 | 65.9 | 19.8 |
| Num. of classes | 2 | 2 | 3 | 2 |

Table 2: Details of Social Network and Bioinformatics datasets.

IMDB-BINARY and IMDB-MULTI are movie collaboration datasets, where each graph corresponds to an ego-network for each actor/actress, nodes corresponds to actors/actresses, and there is an edge between two nodes if they appear in the same movie. Each graph is obtained form a predetermined movie genre, and the task is to classify the graph into the genre from which it is obtained.

COLLAB is a scientific collaboration dataset, obtained from three other collaboration datasets of three different fields: High Energy Physics, Condensed Matter Physics, and Astro Physics. Each graph corresponds to an ego-network of different researchers, and the task is to classify each graph into its corresponding fields.

MUTAG is a dataset of 188 nitroaromatic compounds, where each nodes has 7 discrete labels about their atom types. The goal is to predict their mutagenicity on Salmonella typhimurium.

The descriptor $g_u$ of each node $u$ contains 7 features obtained from the topological properties of $u$ and its neighborhood. They are briefly described here:

1. *Degree:* number of edges
2. *Eccentricity:* On a connected graph, it is the maximum distance between $u$ to all other nodes.
3. *Number of triangles:* The number of triangles having $u$ as a vertex.
4. *Clustering coefficient:* On unweighted graphs, it is the ratio between the number of possible triangles through $u$ versus the number that actually exists.

$$c(u) = \frac{2T(u)}{\deg(u)(\deg(u) - 1)}$$

5. *Core number:* A $k$-core is largest subgraph that only contains nodes of degree at least $k$. The core number of $u$ is the maximum $k$ of all the $k$-core containing $u$.

6. *Clique number:* A clique is a subgraph of an undirected graph such that every two distinct vertices in a clique are adjacent. The clique number of $u$ is the number of cliques containing $u$.

7. *PageRank:* This calculate the PageRank of $u$, which is a ranking of the nodes in the graph based on the topology of the edges.

**Experiments:**

We implemented both AGSN+MLP and GSN+MLP using PyTorch (Paszke et al. (2019)) and PyTorch Geometric (Fey & Lenssen (2019)), and hyperparameters are tuned using W&B sweep (Biewald (2020)). The experiments are run on an HPC instance with 8 CPU cores, 256GB of RAM, and an NVIDIA A100 40GB GPU. For our experiments, we chose a scattering architecture of $m = 2$ on MUTAG, IMDB-BINARY, and IMDB-MULTI, while on COLLAB we chose $m = 4$. At each layer, the number of wavelets are chosen from the range $[3, 4, 5]$. $k_1$ is chosen from the interval $[1, 5]$, while $k_2$ is chosen from $[0.05, 0.15]$. We chose ReLU as the nonlinearity. The architecture is then applied to each signal channel independently, and afterward concatenated to one final representation vector for each graph. As the scattering vector has unequal energy distribution between its path variables (Figure 1) (Bruna & Mallat (2013)), we grouped variables across channels into subvectors and normalized these subvectors using the maximum ($l^2$-norm) across all samples in the batch to enhance robustness:

$$\frac{\phi_k[p](x_i)}{\sup_{x_i} ||\phi_k[p]x_i||}$$

This representation was then directly used as input to the 3-layer MLP for classification.

For temporal discretization, we use Euler schemes, as we found out that the performance gain from using the Runge-Kutta method was negligible. The time-step $\tau$ in the adaptive model is chosen to be 0.25, as lower values resulted in diminishing returns while increasing computational overhead. The number of heads was set to 8, while embedding dimension is chosen from $[16, 32, 64, 128]$. We trained the model for 500 epochs, selected the learning rate from the interval $[0.005, 0.02]$ and applied an exponential decay that halved the learning rate every 50 epochs. The batch size was set to the maximum capacity supported by our hardware. We also applied dropout to MLP layers, with $p$ of dropout chosen from the interval $[0, 0.9]$.

In the first experiment on MUTAG and IMDB-BINARY, we performed $k$-fold cross-validation where $k$ depends on the percentage of data used for training, if the percentage is $\geq 10$. When it is smaller, we use stratified split, with 20 random splits. As the datasets are small, we took the average validation accuracy for each epoch, and chose the best-performing epoch, following Xu et al. (2018a).

In our second experiment on all 4 datasets in Appendix A.6, we perform 10-fold cross-validation, taking average of accuracies across folds similar to above. In both cases, the mean and standard deviation were computed using the best-performing epoch.

