# OpenReview forum: "Graph Scattering Networks with Adaptive Diffusion Kernels"
_ICLR.cc/2025/Conference — Submitted to ICLR 2025_

### Official Review · Reviewer_PdCQ · 2024-10-31

**Soundness:** 2
**Presentation:** 3
**Contribution:** 2
**Rating:** 3
**Confidence:** 4

**Summary:**

This paper studies graph scattering network, and develops an adaptive kernels in diffusion wavelets. The authors further analyze its stability. The experiments show the general improvements over fixed kernel based diffusion wavelets.

**Strengths:**

1. As the authors mention, most of current scattering networks utilize fixed filter banks. Adaptability is a direction to improve them.
2. This paper is rigorous, giving strict definitions and theorems to support arguments.

**Weaknesses:**

1. Motivation conflict. The authors firstly acknowledged scattering network are advantageous with limited data availability at Line 31-33 because of no required training. However, the main motivation of this paper is to make existing scattering learnable and adaptive, which scarifies the internal advantages mentioned above.
2. Complexity. For a scattering network with $L$ layer and $h$ children for each parent node, the total number of filters is $\sum_{l=1}^{L}h^l$, an exponential function. If we make all filters learnable, the computing is very high and unbearable.
3. Experiments. The baselines are too old. More graph scattering methods are suggested to compare.
4. Writing. Starting from section 4, all following equations do not have a mark.

**Questions:**

The authors prove propositions 4.1 and 4.2, Lemma 4.1 and Theorem 4.1. However, only proportion 4.1 describes the proposed adaptive kernel, with language, while the other three prove something for general diffusion wavelet. Therefore, I am not sure their values for this paper, and if these general conclusions have been similarly proposed by previous papers.

---

> ### Author Response · Authors · 2024-11-25
> **Rebuttal by Authors**
>
> We thank you for your constructive feedback. We deeply appreciate the time and effort you invested in evaluating our work. We are pleased to see that you found our approach to be `rigorous` and that it `demonstrates general improvements over fixed kernel-based diffusion wavelets`. We address the reviewer's concern and questions in the following.
>
> > Motivation conflict. The authors firstly acknowledged scattering network are advantageous with limited data availability at Line 31-33 because of no required training. However, the main motivation of this paper is to make existing scattering learnable and adaptive, which scarifies the internal advantages mentioned above.
>
> Thank you for raising this point. In the revised manuscript, we have made changes to clarify more our motivation. In classification tasks, scattering transform is used as a feature extractor, which is used to return a representation usually for classification. Training is only required for the classifier. Since the scattering network are traditionally built from the same wavelet decomposition operator, which in turn is built from a single mother wavelet, directly learning the only corresponding kernel could add negligible number of learnable parameters, comparing to the one that we already have to learn. This can bring much more performance to the model, given the importance of kernel choice. We are sorry for any confusion.
>
> > Complexity. For a scattering network with $L$ layer and $h$ children for each parent node, the total number of filters is $\sum_{l=1}^Lh^l$, an exponential function. If we make all filters learnable, the computing is very high and unbearable.
>
> Thank you for raising this point. As we have discussed above, only one kernel needs to be learned, so computational costs are manageable. We have addressed this in the newly added subsection 4.5 of the revised manuscript, which analyzes the model complexity.
>
> > Experiments. The baselines are too old. More graph scattering methods are suggested to compare.
>
> As we employed adaptive kernel to scattering networks, we compare it to other kernel methods, which are a little bit old. We refer the reviewer to the revised Section 6 of the updated manuscript, where we have added more experiments to directly compare AGSN+MLP to GSN+MLP and GS-SVM [1] and other graph deep learning methods in data scarce scenario, also to address reviewer vroP's concern. We did not employed GGSN+EK [2] as mentioned by reviewer vroP, since the original implementation is not publicly available.
>
> > Writing. Starting from section 4, all following equations do not have a mark.
>
> We are sorry for this oversight. We have add marks to all equations in the revised manuscript.
>
> > The authors prove propositions 4.1 and 4.2, Lemma 4.1 and Theorem 4.1. However, only proportion 4.1 describes the proposed adaptive kernel, with language, while the other three prove something for general diffusion wavelet. Therefore, I am not sure their values for this paper, and if these general conclusions have been similarly proposed by previous papers.
>
> Thank you for your question. In our paper, following the general formulation of the diffusion wavelets in [3], we extends the stability analysis of graph scattering transform to cases where the diffusion operator $A$ is not self-adjoint. And we give a specific design of an adaptive kernel to demonstrate the enhanced performance of adaptivity in scattering network. To our knowledge, there are conclusions which are similar in form have been proposed by previous works, but their proofs assume self-adjointness in $A$, and this is a strong assumption, given the spectral theorem. This is one of the main results of our paper. We believe, in a sense, that our results help complete the picture, and lay the groundwork for new adaptive kernel designs, and applications of non-self-adjoint diffusion operators in scattering networks. To highlight this contribution, we have revised Section 1,4 and 7 for clarity.
> ____
>
> [1] Feng Gao, Guy Wolf, and Matthew Hirn. Geometric scattering for graph data analysis. In International Conference on Machine Learning, pp. 2122–2131. PMLR, 2019. \
> [2] Christian Koke and Gitta Kutyniok. Graph scattering beyond wavelet shackles. In Sanmi Koyejo, S. Mohamed, A. Agarwal, Danielle Belgrave, K. Cho, and A. Oh (eds.), Advances in Neural Information Processing Systems 35: Annual Conference on Neural Information Processing Systems 2022, NeurIPS 2022, New Orleans, LA, USA, November 28 - December 9, 2022, 2022. \
> [3] Ronald R. Coifman and Mauro Maggioni. Diffusion wavelets. Applied and Computational Harmonic Analysis, 21(1):53–94, 2006. ISSN 1063-5203. Special Issue: Diffusion Maps and Wavelets.

---

### Official Review · Reviewer_5wJQ · 2024-11-01

**Soundness:** 3
**Presentation:** 3
**Contribution:** 3
**Rating:** 6
**Confidence:** 3

**Summary:**

This paper seeks to develop a generalized graph scattering transform which learns the transition matrix $A$ through a kernel inspired by the attentional diffusion method from Chamberlain et al. (2021).

They take the initial node features $g_u$, map them into an embedding space by a learnable function $W$ and then build a diffusion operator via a kernel derived from the \{W(g_u)\}_{u\in V} and further add in learning of the diffusion operator via a multiheaded attention mechanism.

After the attention mechanism, they then use the diffusion matrix $A$ to define diffusion wavelets of the form $\psi_j=A^{t_{j-1}}-A^{t_j}$ and use these wavelets to define a graph scattering transform. (This part is ``standard” and similar to other works such as Gama et al. (2019a) and Gao et al. (2019).) Additionally, they prove that their generalized graph scattering transform has similar theoretical properties to other versions of the geometric scattering transform and show strong numerical performance.

Overall, I think this is a good paper which needs a bit of work before it is publication worthy as described below. If these concerns are sufficiently addressed, I will likely raise my score.

**Strengths:**

The geometric scattering transform (GST) provides a theoretically solid framework for understanding multi-scale GNNs from a graph signal processing point of view. However, the original versions of it are limited in their numerical effectiveness because they are overly handcrafted. This paper shows viable ways of increasing the effectiveness of the (GST) while retaining its nice theoretical properties. This therefore helps bridge the gap between ``things that work well” and ``things which are well understood” which is important since GNNs etc are increasingly used in real-world tasks.

**Weaknesses:**

Right before the start of Section 4, I think $v$ should be defined in terms of the square-root of the degree vector (since you are using the symmetrized diffusion operator).

The discussion of Forward Euler etc in the end of Section 4.1 seems out of place in this paper. While it is indeed a useful insight from GRAND etc., I don’t see its relevance on diffusion wavelets which are already in discrete time

It seems to me that you should be able to take $N(\beta_A)=1$ 1 in Proposition 4.2 by imitating the proof of Proposition 4.1 of Gama et al. (2019a). (It might also be useful to look at the proof of Proposition 2.2 of Perlmutter et al `` Understanding Graph Neural Networks with Generalized Geometric Scattering Transforms” (2023).) I believe this would then allow you to establish the stability your method to additive noise (as is common in most formulations of the scattering transform).

Related Works:

Important: The second paragraph omits `` Graph Convolutional Neural Networks via Scattering” (Zou and Lerman 2020). This omission is particularly noteworthy because it is the first paper on graph scattering, predating Gama et al. by a couple of months. (The final publication date is later, but this is an artifact of the journal review process.)

Less important: Additionally, the discussion of incorporating learning into geometric scattering (Section 5) should likely also include ``Overcoming Oversmoothness in Graph Convolutional Networks via Hybrid Scattering Networks” (Wenkel et al. 2022). This paper introduced learning into the scattering framework in a different way than the Tong et al. paper that the authors mention. (As noted in Tong, these two forms of learnable scattering, as well as this one, are compatible and can be combined.) It also should likely include `` Scattering Networks for Hybrid Representation Learning” (Oyallon et al. 2018) and `` Separation and Concentration in Deep Networks” (Zarka et al. 2020) which incorporate learning into Euclidean scattering.

Notational inconsistencies:

There is inconsistent use of ``x” vs ``u” in Section 4.1

$A^*$ is used without being defined. Also, why do you use both $^*$ and $^T$ in equation 2? If the matrices are real this should be the same, right?

In line 221, it would be more natural to call $p_\epsilon$ instead $d_\epsilon to be consistent with the proceed paragraph (or instead call them both $p$)

Very Minor: (Do not affect my score but should be fixed)

Line 54: ``we pursue on alleviating” is awkward. Please rephrase.
Throughout: Things like ``Section 5” are proper nouns and should be capitalized.
Throughout: Some quotation marks point the wrong way (which is an unfortunate artifact of LaTeX sometimes being a pain)

Throughout, some of the equations with $e^{stuff}$ are hard to read and it would be better to write $\exp(stuff)$.

**Questions:**

Why do we need a metric space, in addition to the measure space? Chew et al. `` Geometric scattering on measure spaces” (2024) should that the geometric scattering transform could be extended to measure spaces where there was not an obvious metric structure (e.g., digraphs).

In Section 3.1, what norm is being used? This is relevant because $P$ is an asymmetric matrix (which is similar to \overline{P} used in Section 3.2). In Perlmutter et al `` Understanding Graph Neural Networks with Generalized Geometric Scattering Transforms” (2023) it was shown that the natural choice of norm differs depending on whether one uses a symmetric or asymmetric diffusion matrix (for asymmetric matrices you should use a norm weighted in terms of degrees).

In Section 3.1, could you provide examples of when there we be a strict inequality in line 140? It seems to me that in most cases, this will be equality. For the standard diffusion matrix, I believe that the bottom eigenspace will be zeroed out in the limit, but the other eigenspaces are never fully suppressed (which is why you need epsilon in the next paragraph).

Could the authors please provide examples of setting where having dependence on the sampling density is preferable (as claimed in line 226, ``Naturally…”)? I thought this was typically a nuisance which one would attempt to normalize out (as in line 222).

The unnumbered equations in lines 246-251 are hard to understand. Could the authors please provide some intuition?

---

> ### Author Response · Authors · 2024-11-25
> **Rebuttal by Authors (1/3)**
>
> We thank you for your constructive feedback. We deeply appreciate the time and effort you invested in evaluating our work. We are pleased to see that you consider this a `good paper` that `helps bridge the gap between things that work well and things which are well understood`. We address the reviewer's concerns and questions in the following.
>
> > Right before the start of Section 4, $v$ should be defined in terms of the square-root of the degree vector.
>
> We are really sorry for this typo. We intended it to be $(\frac{\mathbf{d}}{||\mathbf{d}||_1})^{1/2}$. This has been fixed in the revised manuscript.
>
> > The discussion of Forward Euler etc in the end of Section 4.1 seems out of place in this paper.
>
> Here we used the insights taken in [1,2] for adapting continuous diffusion process to graph neural networks. It is true that for traditional diffusion graph scattering networks, this does not change anything, but in our case, using lower time step between each updates or higher order discretization schemes makes the weight update more refined and thus the training process more stable, similarly to reducing the learning rate. We have run additional experiments, presented in Appendix A.6, with a pointer in Section 6 to show this. However, due to the highly non-convex nature of the optimization problem in our case, this does not necessarily translate to better performance, as observed in [1] for linear GCNs. A balance should be achieved between stability and the ability to escape local minima. So we use it as a hyperparameter in our experiments.
>
> To maintain focus, we have also moved the detailed discussion of temporal discretization schemes to Appendix A.1. We refer the reviewer to the revised manuscript for these changes, along with the new experiments.
>
> > It seems to me that you should be able to take $N(\beta_A) = 1$ in Proposition 4.2 by imitating the proof of Proposition 4.1 of Gama et al. (2019a). (It might also be useful to look at the proof of Proposition 2.2 of Perlmutter et al `` Understanding Graph Neural Networks with Generalized Geometric Scattering Transforms” (2023).) I believe this would then allow you to establish the stability your method to additive noise (as is common in most formulations of the scattering transform).
>
> For self-adjoint diffusion operators, the proof in [3,4] indeed allows $N(\beta_A)=1$. This can also be obtain in the presented proof as the case $a \geq 0$ in Appendix A.3. However, in the formulation of diffusion wavelets in [5], the operator can be more general: they may not be self-adjoint. Our proof extends the stability results to this broader case, which is one of the main results of our paper. To address potential misunderstandings, we have revised Sections 1, 4, and 7 to clarify our contributions. We apologize for any confusion caused in the earlier version.
>
> > Related Works
>
> We truly appreciate your suggestions. We have fixed the related works sections in our revised manuscripts for a more accurate and better presentation of prior works.
>
> > Notational inconsistencies and presentation issues. \
> $A^*$ is used without being defined. Also, why do you use both and $^*$ and $^T$ in equation 2? If the matrices are real this should be the same, right?
>
> We are really sorry for these mistakes and typos. We have made the needed changes in the revised manuscript. About $^*$ and $^T$, they are numerically the same in the real case, but we intended different meaning for each. $(A^t_G)^*(A^t_G)$ is the Gram matrix of the vector system $\{A^t_G\delta_u\}_{u\in V}$ where $V$ is the node set of $G$.  On the other hand, $\Pi^{T}$ is actually equal to $\Pi^{-1}$ for a permutation matrix $\Pi$. The formula in equation 2 thus compare the 2 vector systems intrinsic to each graphs $G$ and $G'$, invariant of rotation, translation, or any node permutation. But you are right, using both can cause potential confusion. We have fixed the formula in the revised version.

---

> > ### Author Response · Authors · 2024-11-25
> > **Rebuttal by Authors (2/3)**
> >
> > > Why do we need a metric space, in addition to the measure space? Chew et al. “ Geometric scattering on measure spaces” (2024) should that the geometric scattering transform could be extended to measure spaces where there was not an obvious metric structure (e.g., digraphs).
> >
> > Thank you for the question. The diffusion graph scattering formulation in our paper is constructed on an underlying metric measure space, whereas [5] shows that a graph scattering network can be built without relying on a metric. We believe this is essentially in the differences of how the wavelets used in each scattering transform are constructed, rather than the nature of the architectures themselves.
> >
> > Wavelets are functions that are localized in both spatial and frequential domains. In [6], the author defines the wavelets based on the locality on the spatial domain, then it becomes similar to classical wavelets on $\mathbb{R}^d$. The wavelets in [5] seems to be built from spectral wavelets ([7]), where they are first defined based on the locality on frequential domain, which mostly only rely on the operator  $\mathcal{L}$ being self-adjoint and positive semi-definite. However, as also shown in [7], to analyze the wavelets on the spatial domain, the notion of "locality" first need to be defined, which in turn also necessitates a metric on the space.
> >
> > > In Section 3.1, what norm is being used? This is relevant because P is an asymmetric matrix (which is similar to P used in Section 3.2). In Perlmutter et al “ Understanding Graph Neural Networks with Generalized Geometric Scattering Transforms” (2023) it was shown that the natural choice of norm differs depending on whether one uses a symmetric or asymmetric diffusion matrix (for asymmetric matrices you should use a norm weighted in terms of degrees).
> >
> > Thank you for your question. You are correct that $P$ is self-adjoint with respect to the inner product $\langle f,g \rangle = \sum_{u} \mathbf{v}(u)f(u)g(u)$ where $\mathbf{v}$ is the normalized degree vector. This helps with the analysis of $P$, as shown in [4]. However, in Section 3.1, as presented in the paper, the wavelets construction does not require $P$ to be self-adjoint. It only requires $P$ to be positivity-preserving (which holds by definition of $P$) and contractive (which is satisfied in the example of $(V,E,W)$ that we consider).
> >
> > > In Section 3.1, could you provide examples of when there we be a strict inequality in line 140? It seems to me that in most cases, this will be equality. For the standard diffusion matrix, I believe that the bottom eigenspace will be zeroed out in the limit, but the other eigenspaces are never fully suppressed (which is why you need epsilon in the next paragraph).
> >
> > Thank you for your question. In case where all the eigenvalues of $A$ has moduli larger than $\epsilon$, then yes, the inequality $\dim(A\mathcal{L}^2(X)) \leq \dim{\mathcal{L}^2(X)}$ will not be strict. However, we would like to point out that, by Perron-Frobenius theory, all eigenvalues except $1$ of a stochastic matrix $A$ have their moduli strictly smaller than $1$. Thus all eigenspaces other than the stationary distribution will be zeroed out in the limit of $A^t$ as $t$ tends to $\infty$. We also would like to point out that we in fact defined the numerical accuracy $\epsilon$ before the inequality in line 140, and if the operator $A$ has its smallest eigenvalue (in terms of modulus) smaller than $\epsilon$, then the inequality will also be strict. But it is in fact not that clear in the previous manuscript, and we have made changes to the revised version for clarification.

---

> > > ### Author Response · Authors · 2024-11-25
> > > **Rebuttal by Authors (3/3)**
> > >
> > > > Could the authors please provide examples of setting where having dependence on the sampling density is preferable
> > >
> > > Thank you for your question. For examples, we refer the reviewers to [8], where the authors construct an affinity matrix with Laplacian normalization on the kernel $\exp(-||x-y||^2/\epsilon)$ that we discussed. This is used for image segmentation using spectral clustering. The algorithm presented there used the second eigenvector (ranked in terms of magnitude of the corresponding eigenvalue) of this matrix, and it effectively distinguish clusters with largely different numbers of samples. Another application of this matrix can be found in [9].
> > >
> > > > The unnumbered equations in lines 246-251 are hard to understand. Could the authors please provide some intuition?
> > >
> > > Thank you for the question. There is a typo in the original equation, which we have now fixed. We are sorry for this, and please refer to the revised manuscript for the updated one. We provide some intuition for the approach as follow:
> > >
> > > For two different nodes $u$ and $v$, the diffusion weight between the two is calculated with the usual Laplacian normalization ($k(u,v)/\sum_{v'\in\mathcal{N}(u)}k(u,v')$). However, for adding self-diffusion, $k$ can not be directly used, as the cosine similarity between 2 equal vectors is always 1. Thus, we added a learnable scaling function $\beta(u) = \sigma(\alpha(u))*(1-k_2) + k_2/2$ that determine how much self-diffusion should be given to each node $u$. The diffusion weights of all other neighbor nodes $v$ is then rescaled by $\beta(u)$, keeping $1-\beta(u)$ of the diffusion mass in $u$. The hyperparameter $k_2$ is used to control the range of this $\beta$, similarly to $k_1$.
> > > ______
> > >
> > > [1] Yifei Wang, Yisen Wang, Jiansheng Yang, and Zhouchen Lin. Dissecting the diffusion process in linear graph convolutional networks. Advances in Neural Information Processing Systems, 34: 5758–5769, 2021.\
> > > [2] Ben Chamberlain, James Rowbottom, Maria I Gorinova, Michael Bronstein, Stefan Webb, and Emanuele Rossi. Grand: Graph neural diffusion. In International Conference on Machine Learning, pp. 1407–1418. PMLR, 2021 \
> > > [3] Fernando Gama, Alejandro Ribeiro, and Joan Bruna. Diffusion scattering transforms on graphs. In International Conference on Learning Representations, 2019a.\
> > > [4] Perlmutter, M., Tong, A., Gao, F., Wolf, G., \& Hirn, M. (2023). Understanding graph neural networks with generalized geometric scattering transforms. SIAM Journal on Mathematics of Data Science, 5(4), 873-898.\
> > > [5] Chew, Joyce, et al. "Geometric scattering on measure spaces." arXiv preprint arXiv:2208.08561 (2022). \
> > > [6] Ronald R. Coifman and Mauro Maggioni. Diffusion wavelets. Applied and Computational Harmonic Analysis, 21(1):53–94, 2006. ISSN 1063-5203. Special Issue: Diffusion Maps and Wavelets.\
> > > [7] David K. Hammond, Pierre Vandergheynst, and R´emi Gribonval. Wavelets on graphs via spectral graph theory. Applied and Computational Harmonic Analysis, 30(2):129–150, 2011. ISSN 1063-5203\
> > > [8] Weiss, Yair. "Segmentation using eigenvectors: a unifying view." Proceedings of the seventh IEEE international conference on computer vision. Vol. 2. IEEE, 1999.\
> > > [9] Shi, Jianbo, and Jitendra Malik. "Normalized cuts and image segmentation." IEEE Transactions on pattern analysis and machine intelligence 22.8 (2000): 888-905.

---

> > > > ### Comment · Reviewer_5wJQ · 2024-11-26
> > > > **Raised Score**
> > > >
> > > > Thank you for your thorough response. I have increased my score to a six

---

> > > > > ### Author Response · Authors · 2024-11-26
> > > > > **Thank you**
> > > > >
> > > > > Thank you for taking a careful look at our rebuttal and raising the score. We are glad that our rebuttal has alleviated your concerns.

---

### Official Review · Reviewer_pJh1 · 2024-11-03

**Soundness:** 2
**Presentation:** 2
**Contribution:** 1
**Rating:** 3
**Confidence:** 3

**Summary:**

This work introduces a method for incorporating adaptive kernels into graph scattering networks. The paper provides theoretical stability guarantees against input data perturbations, ensuring robustness. Experimental results demonstrate that adaptive kernels offer advantages over traditional scattering networks.

**Strengths:**

1. The theoretical analysis to support the advantages of the adaptive wavelet diffusion.

**Weaknesses:**

1. The authors highlight the limitations of traditional methods under low-data scenarios. However, the paper lacks theoretical analysis or specific experiments tailored to illustrate how the proposed adaptive kernel-based scattering networks (AGSN) address performance in data-scarce environments.

2. The experimental results reveal that AGSN does not outperform some well-known graph classification techniques.

3. The limitation of The Related Works. There are some works also related to adaptive kernels for graph neural networks, such as [1-2]. It is not clear the advantages of the proposed adaptive wavelet diffusion compared to others.

[1] Sun, C., Hu, J., Gu, H., Chen, J. and Yang, M., 2020. Adaptive graph diffusion networks. arXiv preprint arXiv:2012.15024.

[2] Zhao, J., Dong, Y., Ding, M., Kharlamov, E. and Tang, J., 2021. Adaptive diffusion in graph neural networks. Advances in neural information processing systems.

4. The paper lacks the complexity analysis.

**Questions:**

What are the advantages of the proposed adaptive wavelet diffusion compared to other methods [1-2] ?

---

> ### Author Response · Authors · 2024-11-25
> **Rebuttal by Authors**
>
> We sincerely thank you for your constructive feedback and for the time and effort you invested in evaluating our work. We are pleased that you recognize how the work `demonstrates that adaptive kernels offer advantages over traditional scattering networks`. We address the reviewer's concerns and questions in the following.
>
> > The authors highlight the limitations of traditional methods under low-data scenarios. However, the paper lacks theoretical analysis or specific experiments tailored to illustrate how the proposed adaptive kernel-based scattering networks (AGSN) address performance in data-scarce environments.
>
> We appreciate the reviewer's observation regarding the need for more analysis in low-data scenarios. In the original version, we compared AGSN+MLP and GSN+MLP under varying data conditions, ranging from 10\% to 90\% of the data used for training. The results show AGSN consistently outperform GSN. To address this and reviewer vroP's concerns, we have also conducted additional experiments in even scarcer data scenarios and included comparisons with other graph deep learning and scattering methods. These additions are now detailed in Section 6 of the revised manuscript.
>
> > The experimental results reveal that AGSN does not outperform some well-known graph classification techniques.
>
> Thank you for raising this point. To address this, we have conducted additional experiments comparing AGSN+MLP with both traditional graph scattering methods and graph deep learning techniques (e.g., UGformer and GIN-0) in data-scarce environments. These new results demonstrate that AGSN+MLP outperforms these methods in such settings. Additionally, we clarify that the results in Table 1 of the original paper are included to provide context for abundant training data scenarios and are not the primary focus of our contributions. To provide space for addressing the reviewers' concerns and to maintain the focus of the paper, we have moved it to Appendix A.6 in the revised version.
>
> > The limitation of The Related Works. There are some works also related to adaptive kernels for graph neural networks, such as [1-2]. It is not clear the advantages of the proposed adaptive wavelet diffusion compared to others. \
> What are the advantages of the proposed adaptive wavelet diffusion compared to other methods [1-2] ?
>
> Thank you for your question and for pointing to these works. We would like to point out that, while the names of these methods and our adaptive wavelet diffusion are similar, their structures and purposes differ significantly:
>
> + Both [1] and [2] focus on traditional graph neural networks, where the output is returned at the final layer. At each layer, a complete diffusion process using a fixed operator with $K$ steps is done. [1] proposed adaptive diffusion, based on existing GNNs that use multi-hop information in each layer. The output of each layer in previous models is both the sum of the multi-hop information taken via normalized adjacency matrix $\bar{A}$ and also the linear residual connection. The author proposed to make the sum of multi-hop information a weighted sum, and the weights are learnable. [2] instead use the heat kernel $e^{-\mathbf{L}t} = \sum_{k=0}^{\infty}e^{-t}T^k$, where $\mathbf{L}=I+T$ is the Laplacian, as the multi-hop information, and they proposed to adaptively learn $t$ at each layer. They also introduced the generalized version, which is a weighted sum of $\{T^i\}_i$, where the weights are also learnable. These two methods both use fixed operator for the diffusion process.
>
> + Our proposed method, on the other hand, makes the diffusion operator itself learnable, enabling the construction of adaptive diffusion wavelets. The wavelets are inherently designed to capture multi-scale information [3]. In the scattering transform, information at multiple scales is concatenated and used for other tasks. Our approach emphasizes more about learning the diffusion process, rather than learning the aggregation of information from a predefined one.
>
> > The paper lacks the complexity analysis.
>
> Thank you for raising this concern. In the revised manuscript, we have added Subsection 4.5, which provides a detailed complexity analysis of AGSN. Additionally, we have included runtime experiments in Section 6 to address this point, as well as related concerns raised by reviewers zGfS and PdCQ.
>
> ___
>
> [1] Sun, C., Hu, J., Gu, H., Chen, J. and Yang, M., 2020. Adaptive graph diffusion networks. arXiv preprint arXiv:2012.15024. \
> [2] Zhao, J., Dong, Y., Ding, M., Kharlamov, E. and Tang, J., 2021. Adaptive diffusion in graph neural networks. Advances in neural information processing systems.\
> [3] Ronald R. Coifman and Mauro Maggioni. Diffusion wavelets. Applied and Computational Harmonic Analysis, 21(1):53–94, 2006. ISSN 1063-5203. Special Issue: Diffusion Maps and Wavelets.

---

### Official Review · Reviewer_vroP · 2024-11-03

**Soundness:** 2
**Presentation:** 2
**Contribution:** 2
**Rating:** 5
**Confidence:** 3

**Summary:**

This paper proposes a mathematically sound framework for applying adaptive kernels to diffusion wavelets, thus overcoming the limitations of traditional graph scattering networks with predefined wavelets.

**Strengths:**

* Considering the importance of selecting an appropriate kernel, it is promising to develop a framework for application of adaptive kernels in graph scattering networks.

* The proposed framework is bulit on mathematically sound foundation, and stability guarantees with respect to input perturbations are also provided, thus enhanceing its rationality and reliability.

* The experimental results also demonstrated that it consistently outperforms traditional graph scattering networks.

**Weaknesses:**

The main problem with this paper is that its experiments are not convincing enough.

* Baselines:
    * Given that graph deep learning has developed rapidly in recent years, this paper lacks comparisons against the latest graph deep learning methods.
    * More importantly, some typical graph scattering transform methods are not employed and compared in the experiments, such as GS-SVM [1] and GGSN+EK [2].

* The experimental results can not support the clained superiority.
Although the authors have given some explanations, why not further conduct some experiments to prove it?
For example, it's necessary to report the performance of deep learning methods when low training-data availability to prove the meaning of this work.

[1] Gao F, Wolf G, Hirn M. Geometric scattering for graph data analysis[C]//International Conference on Machine Learning. PMLR, 2019: 2122-2131.

[2] Koke C, Kutyniok G. Graph scattering beyond wavelet shackles[J]. Advances in Neural Information Processing Systems, 2022, 35: 30219-30232.

**Questions:**

Please see the **weaknesses** part.

---

> ### Author Response · Authors · 2024-11-25
> **Rebuttal by Authors**
>
> We thank you for your constructive feedback. We deeply appreciate the time and effort you invested in evaluating our work. We are pleased to see that you find our work `promising`, `enhancing rationality and reliability`, and `demonstrating consistent outperformance over traditional graph scattering methods`. We address the reviewer's concerns in the following.
>
> >The main problem with this paper is that its experiments are not convincing enough: \
>     + Baselines. \
>     + It's necessary to report the performance of deep learning methods when low training-data availability.
>
> In Section 6, we have included additional experiments, comparing AGSN+MLP with GSN+MLP and graph deep learning methods (graph transformer (UGformer [1]), graph neural network (GIN-0 (MLP-sum)[2])), and another graph scattering method (GS-SVM [3]) in scenarios with low training data availability. We also added runtime experiments to address reviewer zGfS and pJh1 questions and concerns.
>
> We did not include GGSN+EK [4] in our experiments as the original implementation is not publicly available. However, for completeness, we have added the results presented in [4] for comparison in Table 1, which was initially included to compare the models in scenarios with abundant training data, and is now moved to the additional experiments section in Appendix A.6 to maintain the focus of the main paper.
> ___
>
> [1] Dai Quoc Nguyen, Tu Dinh Nguyen, and Dinh Phung. Universal graph transformer self-attention networks. In Companion Proceedings of the Web Conference 2022, pp. 193–196, 2022. \
> [2] Keyulu Xu, Weihua Hu, Jure Leskovec, and Stefanie Jegelka. How powerful are graph neural networks? In International Conference on Learning Representations, 2018. \
> [3] Feng Gao, Guy Wolf, and Matthew Hirn. Geometric scattering for graph data analysis. In International Conference on Machine Learning, pp. 2122–2131. PMLR, 2019. \
> [4] Christian Koke and Gitta Kutyniok. Graph scattering beyond wavelet shackles. In Sanmi Koyejo, S. Mohamed, A. Agarwal, Danielle Belgrave, K. Cho, and A. Oh (eds.), Advances in Neural Information Processing Systems 35: Annual Conference on Neural Information Processing Systems 2022, NeurIPS 2022, New Orleans, LA, USA, November 28 - December 9, 2022, 2022

---

### Official Review · Reviewer_zGfS · 2024-11-04

**Soundness:** 2
**Presentation:** 3
**Contribution:** 1
**Rating:** 5
**Confidence:** 4

**Summary:**

This paper introduces Graph Scattering Networks with Adaptive Diffusion Kernel, which enhances traditional graph scattering networks by incorporating learnable kernels while maintaining mathematical soundness. The novel part is that it bridges the gap between fixed wavelet transforms and learnable architectures while preserving mathematical guarantees.

**Strengths:**

1. the authors propose a novel framework that  incorporate learnable kernels in graph scattering networks.
2. prove that the adaptive kernels maintain symmetry and self-adjointness
3. provide stability analysis for learnable kernels

**Weaknesses:**

1. As far as I understand, the adaptive kernel is restricted to self-adjoint operators for mathematical convenience.
2. The weak performance raises questions about whether the theoretical advantages of the approach translate to practical benefits.
3. The fundamental question "Why adaptive scattering?" is not convincingly answered in the paper. The theoretical contribution might be interesting, but its practical necessity and benefits are not well established.

**Questions:**

1. What's the running time compared with other GNN methods, Could you provide runtime comparisons with baselines?
2. In what scenarios does adaptivity help/hurt performance? why we need adaptivity?

---

> ### Author Response · Authors · 2024-11-25
> **Rebuttal by Authors**
>
> Thank you for your constructive feedback. We deeply appreciate the time and effort you invested in evaluating our work. We are glad to see you recognize our work as `novel` and appreciate its efforts to `bridge the gap between fixed wavelet transforms and learnable architectures while preserving mathematical guarantees`. However, we believe there are some misunderstandings of our contributions. We address the reviewer's concerns and questions in the following.
>
> > The adaptive kernel is restricted to self-adjoint operators for mathematical convenience.
>
> We respectfully clarify that our theoretical analysis does not restrict the kernel to be self-adjoint. Our stability bounds are derived for general diffusion operators $A$, which only need to be contractive and positivity-preserving. Consequently, our proofs provide a foundation for scattering transforms constructed from other important diffusion operators that are not self-adjoint, and thus other adaptive designs that contain such operators.
>
> To address the potential misunderstanding, we have revised Sections 1, 4, and 7 to make our contributions in this regard clearer. We are really sorry for any confusion.
>
> > The weak performance raises questions about whether the theoretical advantages of the approach translate to practical benefits.
>
> Our primary aim in this paper is to enhance traditional graph scattering methods in low-data scenarios by applying adaptive diffusion kernels, supported by rigorous mathematical analysis. In our original submission, the experiments comparing AGSN+MLP and GSN+MLP with other graph deep learning methods aimed to demonstrate that graph scattering is still outperformed by graph deep learning when abundant training data is available (e.g., with 90\% of the dataset used for training during cross-validation). It is provided to facilitate a broader view.
>
> Based on additionally reviewers vroP and pJh1's concerns, we have now included additional experiments to directly compare AGSN+MLP with other graph learning methods in low data scenarios. The results confirm a consistent performance gap between AGSN and other models in such scenarios. Additionally, across both abundant and scarce data settings, AGSN+MLP consistently outperform traditional graph scattering (GSN+MLP and GS-SVM) by a considerable margin. These findings have been added to the revised manuscript, detailed in Section 6.
>
> > The fundamental question "Why adaptive scattering?" is not convincingly answered in the paper. The theoretical contribution might be interesting, but its practical necessity and benefits are not well established.
>
> Thank you for pointing this out. To address this and also the concern of reviewer PdCQ, we have revised the manuscript to better establish the motivation for adaptive scattering and its practical relevance. We refer the reviewer to the following sections:
>
> * Section 4.1: Here, we discuss the importance of kernel choices in capturing different dataset properties with examples.
>
> * Revised Section 1: We have emphasized more on the motivation for adaptive scattering as there are minimal additional parameters compared to those already in the classifier.
>
> * Revised Section 6: We have included additional experiments, showing that AGSN consistently outperforms graph deep learning methods in low-data scenarios. These results also demonstrate that incorporating adaptive kernels into wavelets leads to consistent performance improvements over traditional graph scattering methods.
>
> > What's the running time compared with other GNN methods, Could you provide runtime comparisons with baselines?
>
> Thank you for your question. We refer the reviewer to the revised Section 6 of the updated manuscript, where we have added runtime comparisons with baseline methods.
>
> > In what scenarios does adaptivity help/hurt performance? why we need adaptivity?
>
> Thank you for this question.
>
> Adaptivity generally enhances performance compared to traditional graph scattering networks, as the representation is usually large enough, making the number of additional parameters relatively small. This is demonstrated in our experiments, where AGSN consistently outperforms its non-adaptive counterpart.
>
> However, there are specific scenarios where adaptivity may not provide significant benefits:
>
> * Extremely low-data scenarios: When the training dataset is exceedingly small (e.g., MUTAG with only 2\% training data, equivalent to ~3 samples), the limited data availability constrains the ability of adaptive kernels to learn effectively. In such cases, adaptive scattering and the traditional version perform similarly.
>
> * Simple datasets: When the data can be effectively classified using a shallow scattering network (e.g., 2 layers with few wavelets), the additional parameters from adaptive kernels may hurt performance due to the relative increase in complexity. In these cases, a traditional scattering network suffices.

---

> > ### Comment · Reviewer_zGfS · 2024-12-03
> > **after rebuttal**
> >
> > Thank you for your response. While I appreciate the clarifications, I remain unconvinced that adaptive scattering provides significant advantages over other methods. Some of my concerns have been addressed, so I have adjusted my score to 5.

---

> > > ### Author Response · Authors · 2024-12-03
> > > **Response by Authors**
> > >
> > > Thank you for taking a careful look at our rebuttal and for raising the score. While it’s true that adaptive scattering doesn’t always provide significant performance advantages over other graph learning methods, in those cases, we see its value in narrowing the performance gap between traditional scattering and deep learning architecture while maintaining both mathematical interpretability and stability guarantees—qualities that, in our view, are significant advantages over many graph deep learning methods that are still considered “black boxes.” We thus believe our work helps lay a foundation for developing graph learning models that are not only more performant but also interpretable in the future.

---

### Author Response · Authors · 2024-11-25
**General Response by Authors**

Dear AC and Reviewers,

We would like to say thanks to all the reviewers for their time and effort to review our work. The thoughtful reviews and constructive feedback help us to improve the paper significantly. We are encouraged to see the recognitions that our work is novel (zGfS), promising (vroP), rigorous (PdCQ), demonstrates general improvements over traditional methods (pJh1, PdCQ), and helps bridge the gap between things that work well and things which are well understood (5wJQ). We have updated the manuscript based on the reviewers' feedback and suggestions.

One of the common concerns is about the experiments and AGSN does not outperform other graph deep learning methods. We would like to address this by clarifying that our main focus of this paper is to improve on traditional graph scattering, which is usually used in low-data scenario as a feature extractor. In our additional experiments in such scenarios, AGSN+MLP shows considerable performance gain compared to both traditional graph scattering (GSN+MLP, GS-SVM) and graph deep learning methods (GIN-0 (MLP-sum), UGformer). We also would like to clarify that the results in Table 1 are provided to show how graph scattering methods are still outperformed by graph deep learning in abundant data scenarios (90\% used for training). These results are added to provide a broader view.

Additionally, there might be some confusion about the restriction on the adaptive kernels, and our theoretical contributions. We would like to clarify that the kernels do not need to be self-adjoint in our theoretical analysis. This thus lays a foundation for scattering networks using important diffusion operator in such class, and also other more general adaptive designs. We have revised our manuscript to make this clearer.

___

We have updated our manuscript to address the comments and questions of the reviewers:
+ Following the comments of reviewers vroP, pJh1, and PdCQ, Section 6 has been updated with additional experiments in low data scenarios with other graph deep learning and scattering methods. We have included other graph scattering methods (GS-SVM, GGSN+EK) to Table 1 for comparison in abundant training data scenarios (suggested by reviewer vroP and PdCQ).
+ Following reviewers zGfS and PdcQ comments, we have revised Sections 1, 4, and 7 to highlight and make clearer our motivations and main contributions in this work.
+ As reviewer 5wTQ pointed out, the discussion on discretization schemes may seem irrelevant. We have included additional experiments in Appendix A.6 to show the effect of schemes and time step choices on the training process and performance of the model, similar to adjusting the learning rate.
+ Following reviewers pJh1, PdCQ, and zGfS comments, we have added the complexity analysis in Section 4.5, and runtime experiments in Section 6.
+ Following reviewers 5wTQ comments, we have added more discussions on other ways of incorporating adaptivity to scattering network, both in graph and Euclidean domains.
+ To maintain the focus of the paper, and leave space to address reviewers' comments, we have moved the discussion on temporal discretization schemes to Appendix A.1, and Table 1 to Appendix A.6 for additional experiments, with a discussion and pointer in Section 6.
+ We have fixed all typos and notational inconsistencies.
___

We are glad to answer any additional question you have on our submission.

Thank you,

Authors

---

### Author Response · Authors · 2024-12-01
**General Response by Authors**

We would like to thank all reviewers once again for their thoughtful feedback and constructive comments.

We would greatly appreciate it if you could let us know if there are additional questions or concerns before the end of the discussion period.

We are happy to engage in any follow-up discussion or address any additional comments.

---

### Meta-Review · Area_Chair_SSXD · 2024-12-18

**Metareview:**

In this submission, the authors proposed a new graph kernel-based learning method with some theoretical guarantees. However, the reviewers and AC have concerns about the inconsistency between the claimed theoretical superiority and the practical performance achieved by the proposed method. Although the proposed method outperforms representative graph kernel methods, it seems inferior to GNN-based competitors in terms of both runtime and accuracy.

In the rebuttal phase, the authors claimed that the proposed method works well when over 90% of data are used for training. However, such a setting is often infeasible in practice. In addition, the datasets (e.g., MUTAG and IMDB-B) considered in this submission are over-simplified. To demonstrate the usefulness of a kernel-inspired graph learning method, it is necessary to test it on large-scale graph datasets. In summary, the authors should enhance the performance of the proposed method, and the submission requires a next-round review.

**Additional Comments On Reviewer Discussion:**

Two reviewers interacted with the authors in the rebuttal phase. In the discussion and decision phases, AC asked for more comments but did not get feedback till Dec. 18. After reading the submission, the comments, and the rebuttals, AC has decided to reject this work.

---

### Decision · Program_Chairs · 2025-01-22

Reject